# TRAF3IP3 negatively regulates cytosolic RNA induced anti-viral signaling by promoting TBK1 K48 ubiquitination

Meng Deng [1,2,3], Jason W. Tam[2,4], Lufei Wang [1], Kaixin Liang[1,2], Sirui Li[2,4], Lu Zhang[2,5], Haitao Guo[2,4], Xiaobo Luo[6], Yang Zhang[7], Alex Petrucelli[2,4], Beckley K. Davis [8], Brian J. Conti[2,9], W. June Brickey [2,10], Ching-Chang Ko[1,3], Yu L. Lei [6], Shaocong Sun [11] & Jenny P. -Y. Ting[1,2,4,10 ✉]

Innate immunity to nucleic acids forms the backbone for anti-viral immunity and several inflammatory diseases. Upon sensing cytosolic viral RNA, retinoic acid-inducible gene-I-like receptors (RLRs) interact with the mitochondrial antiviral signaling protein (MAVS) and activate TANK-binding kinase 1 (TBK1) to induce type I interferon (IFN-I). TRAF3-interacting protein 3 (TRAF3IP3, T3JAM) is essential for T and B cell development. It is also well-expressed by myeloid cells, where its role is unknown. Here we report that TRAF3IP3 suppresses cytosolic poly(I:C), 5'ppp-dsRNA, and vesicular stomatitis virus (VSV) triggers IFN-I expression in overexpression systems and *Traf3ip3*−/− primary myeloid cells. The mechanism of action is through the interaction of TRAF3IP3 with endogenous TRAF3 and TBK1. This leads to the degradative K48 ubiquitination of TBK1 via its K372 residue in a DTX4-dependent fashion. Mice with myeloid-specific gene deletion of *Traf3ip3* have increased RNA virus-triggered IFN-I production and reduced susceptibility to virus. These results identify a function of TRAF3IP3 in the regulation of the host response to cytosolic viral RNA in myeloid cells.

---

[1] Oral and Craniofacial Biomedicine PhD Program, University of North Carolina at Chapel Hill, Chapel Hill, NC 27514, USA. [2] Lineberger Comprehensive Cancer Center, University of North Carolina at Chapel Hill, Chapel Hill, NC 27514, USA. [3] Department of Craniofacial and Surgery Sciences, University of North Carolina at Chapel Hill, Chapel Hill, NC 27514, USA. [4] Department of Genetics, University of North Carolina at Chapel Hill, Chapel Hill, NC 27514, USA. [5] Sarah W. Stedman Nutrition and Metabolism Center and Duke Molecular Physiology Institute, Duke University Medical Center, Durham, NC 27710, USA. [6] Department of Periodontics and Oral Medicine, University of Michigan School of Dentistry, University of Michigan Rogel Cancer Center, University of Michigan, Ann Arbor, MI 48105, USA. [7] Department of Dermatology, the Second Affiliated Hospital, School of Medicine, Xi'an Jiaotong University, Xi'an, Shaanxi 710004, China. [8] Department of Biology, Franklin and Marshall College, Lancaster, PA 17604, USA. [9] Biotechnology Center, University of Wisconsin-Madison, Madison, WI 53706, USA. [10] Department of Microbiology and Immunology, University of North Carolina at Chapel Hill, Chapel Hill, NC 27514, USA. [11] Department of Immunology, The University of Texas MD Anderson Cancer Center, Houston, TX 77030, USA. ✉email: Jenny_ting@med.unc.edu

Innate immunity is the first line of host defense against microbial infection and is evolutionarily conserved in all multicellular organisms across Linnaean borders[1,2]. Host cells utilize germline-encoded receptors, collectively termed pattern recognition receptors (PRRs), to detect conserved microbial components, known as pathogen-associated molecular patterns (PAMPs). Recognition of PAMPs by PRRs activates signaling cascades that lead to the production of an arsenal of effector molecules that restrict microbial invasion. Among these effector molecules, type I interferon (IFN-I) is central to host anti-viral defense by inducing IFN-stimulated genes (ISGs) that contain virus dissemination and activate adaptive immune responses[3,4].

Multiple PRRs have been characterized to surveil the presence of viral nucleic acid and induce IFN-I, including membrane-bound sensors such as Toll-like receptors (TLRs), and cytosolic sensors such as retinoic acid-inducible gene-I (RIG-I) like receptors (RLRs) and cyclic GMP-AMP synthase (cGAS)[5,6]. Cytosolic viral RNA is mainly detected by RIG-I and melanoma differentiation-associated gene 5 (MDA5), both belonging to the DExD/H box RNA helicase family. RIG-I and MDA5 serve nonredundant functions; RIG-I preferentially detects short dsRNA such as synthetic poly(I:C) (~300 bp) and can specifically recognize RNA with 5′-triphosphates, 5′-diphosphates and panhandle structures[7–11], whereas MDA5 recognizes long dsRNA such as poly(I:C) (>4 kb) and RNA devoid of 2′-O-methylation[12,13]. Cytosolic AT-rich dsDNA can also activate RIG-I after transcription into dsRNA by RNA polymerase III[14,15]. Binding of RNA to RIG-I and MDA5 activates the adaptor protein mitochondrial antiviral-signaling (MAVS, also known as IPS-1, CARDIF and VISA). MAVS then assembles a signaling platform that recruits TBK1 to phosphorylate the transcription factor IRF3, leading to the induction of IFN-I signaling[16,17]. Different from cytosolic RNA sensing, endosomal RNA is sensed by TLR3 that recruits the adaptor protein TIR-domain-containing adapter-inducing interferon-β (TRIF) and TLR7 and TLR8 which function via the adaptor, MyD88. TLR3 recognizes dsRNA while TLR7 and TLR8 recognize ssRNA and all three reside in the endosome[6]. Cytosolic DNA is sensed by cGAS that activates the adaptor protein stimulator of interferon genes (STING)[5,6]. Similar to MAVS, the TRIF and STING pathways converge on the recruitment of TBK1 for IFN-I induction[17].

Although activation of IFN-I signaling is important for elimination of invading microbes, inappropriate IFN-I induction is detrimental to immune homeostasis and promotes immunopathology[5,18]. Among the IFN regulatory molecules, TBK1 is crucial for the activation of IRF3 and subsequent IFN-I induction. TBK1 is regulated by posttranslational modifications such as phosphorylation, ubiquitination and acetylation. TBK1 autophosphorylation at Ser172 is essential for its activation[19]. It was reported that GSK3β, PPM1B, PP4 and PPM1A could modulate TBK1 activity by altering the TBK1 phosphorylation state[20–23]. NRDP1 and RNF128 mediate K63-linked polyubiquitination and promotes TBK1 activation[24,25]. NLRP4-DTX4, TRIP and Siglec1-TRIM27 promote K48-linked polyubiquitination that leads to TBK1 degradation[26–29]. HDAC9 deacetylates TBK1 and enhances TBK1 activation[30]. These reports illustrate that TBK1 is under tight multi-layered control but the cellular regulatory mechanisms remain incompletely understood.

TRAF3-interacting protein 3 (TRAF3IP3, also known as TRAF3-interacting JNK-activating modulator (T3JAM)) was initially identified as a TRAF3 interacting protein[31], although significant time lapsed before its physiological function was studied with gene deletion mice. Studies show TRAF3IP3 is required for B and T cell development, as well as for the maintenance of regulatory T cell functional stability[32–34]. By contrast, here we find that TRAF3IP3 is well expressed by cells within the myeloid lineage, where its function in innate immunity is unclear. We establish that TRAF3IP3 is a regulator of the cytosolic RNA-induced IFN-I pathway. TRAF3IP3 associates with components of the cytosolic RNA sensing pathway, including TRAF3 and TBK1 in the maintenance of immune homeostasis and the prevention of overzealous innate immunity.

## Results

**TRAF3IP3 attenuates the type I interferon response.** To study the function of TRAF3IP3, we profiled *TRAF3IP3* gene and TRAF3IP3 protein expression in normal human and murine tissues using several publicly accessible databases including BioGPS, Genecards and Human Protein Atlas. TRAF3IP3 is preferentially expressed in primary and secondary lymphoid organs as well as adaptive and innate immune cells in humans and mice (Supplementary Fig. 1a–d), suggesting immune-specific function of TRAF3IP3. To explore the function of TRAF3IP3 in innate immunity, we investigated whether TRAF3IP3 had a substantial impact on IFN-I signaling. We transfected HEK293T cells with an IFN-β promoter-driven luciferase reporter and internal control *Renilla* luciferase reporter as well as empty vector (EV) or vector encoding TRAF3IP3. Overexpression of TRAF3IP3 did not activate the IFN-β promoter-driven luciferase reporter, indicating TRAF3IP3 is not an activator of IFN-I signaling (Fig. 1a). IFN-β induction requires the coordinated activation of IFN-stimulated response element (ISRE) and NF-κB[35]. We also used an ISRE promoter-driven luciferase reporter or an NF-κB promoter-driven luciferase reporter and found TRAF3IP3 activated neither of these reporters (Fig. 1b and Supplementary Fig. 2a). Therefore, TRAF3IP3 does not activate IFN-I signaling. Instead, we find that TRAF3IP3 reduced IFN-I response. Cytosolic poly(I:C) and 5′ppp-dsRNA stimulation or VSV infection is known to activate MDA5/RIG-I-MAVS dependent IFN-I signaling[8,10,12,36], whereas poly(dA:dT) can activate both RNA sensing RIG-I pathway thorough transcription by RNA polymerase III into RNA[14] and DNA sensing cGAS-STING pathway[37] to induce IFN-I. In HEK293T cells, cytosolic poly(I:C), poly(dA:dT), 5′ppp-dsRNA stimulation or vesicular stomatitis virus (VSV) infection activated IFN-β and ISRE promotor-driven luciferase reporters were all reduced by TRAF3IP3 in a dose-dependent fashion (Fig. 1a–h). To dissect the pathway activated by poly(dA:dT), we performed immunoblotting and found that HEK293T cells did not express detectable endogenous cGAS or STING, albeit HeLa, THP-1 and BJAB cells expressed both, and Jurkat-T cells only expressed STING (Supplementary Fig. 2b). Therefore in HEK293T cells, IFN-β induced by the dsDNA poly(dA:dT) is likely through the RNA polymerase III-directed RIG-I pathway. Activation of IFN-β is associated with IRF3 phosphorylation and translocation from the cytoplasm to the nucleus. IRF3 then binds to ISRE to induce IFN-β. Overexpression of TRAF3IP3 suppressed IRF3 phosphorylation induced by cytosolic poly(I:C), poly(dA:dT) and 5′ppp-dsRNA stimulation (Fig. 1i, j, densitometric measurements shown in Supplementary Fig. 2c, d), and inhibited IRF3 translocation into the nucleus induced by cytosolic poly(I:C) (Fig. 1k). Reporter assay represents an artificial system, thus we next showed that overexpression of TRAF3IP3 also significantly inhibited IFN-β protein secretion induced by cytosolic poly(I:C), poly(dA:dT), 5′ppp-dsRNA stimulation or VSV infection (Fig. 1l). Taken together, these data suggest that TRAF3IP3 inhibits the cytosolic RNA-induced IFN-I pathway.

**Reducing TRAF3IP3 potentiates the type I interferon response.** We next sought to determine the function of endogenous

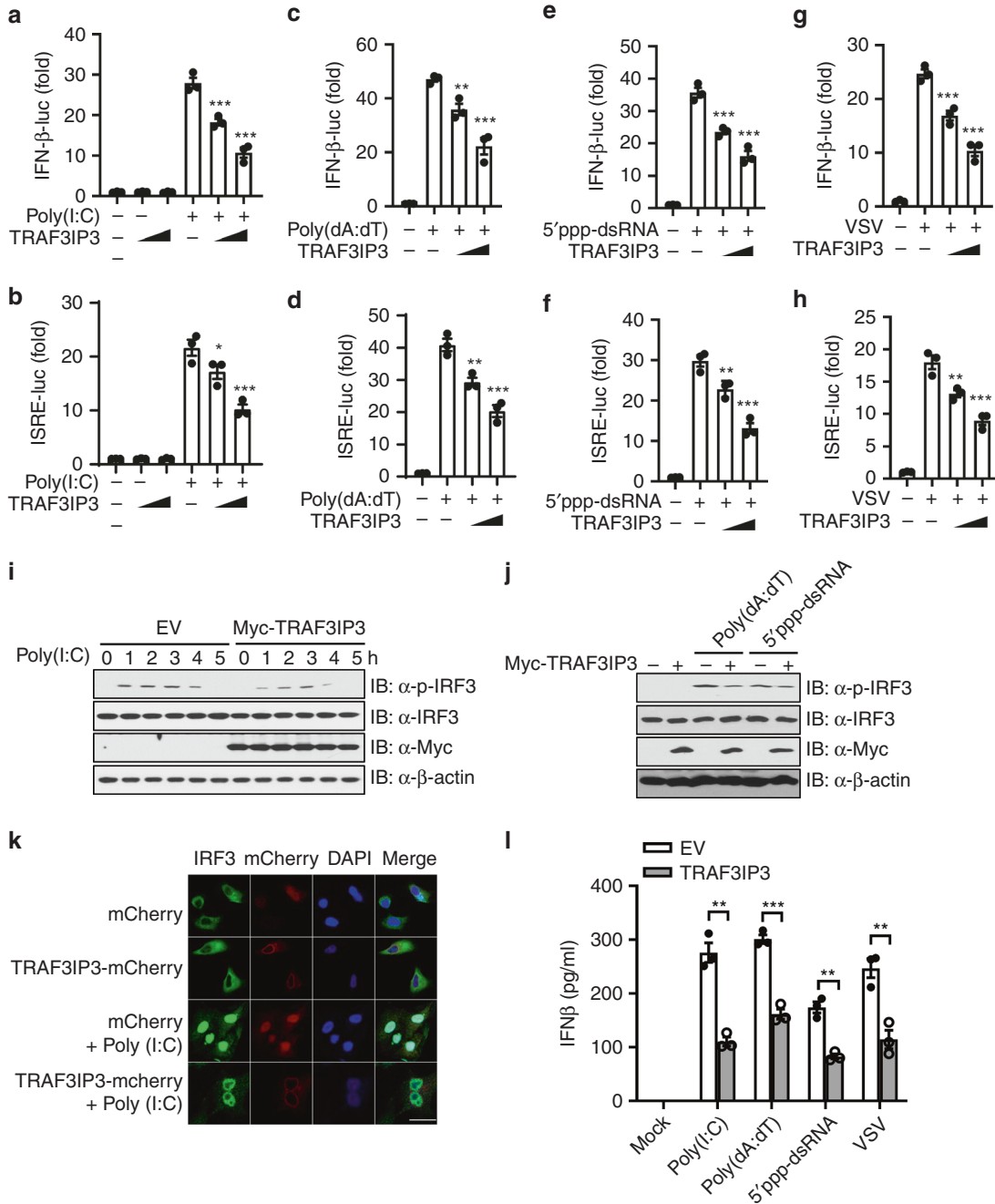

**Fig. 1 TRAF3IP3 attenuates the type I interferon response. a–h** Luciferase assay conducted in HEK293T cells transfected with increasing Myc-TRAF3IP3 (wedge represents 100 and 200 ng) or empty vector (EV), together with the IFN-β or ISRE reporter for 24 h, followed by mock transfection, transfection of poly(I:C), poly(dA:dT), 5′ppp-dsRNA for 6 h or VSV infection (MOI = 0.5) for 6 h. *Renilla* luciferase was used as the internal control. **i, j** Immunoblotting using HEK293T cells transfected with Myc-TRAF3IP3 or empty vector (EV) for 24 h, followed by mock transfection, transfection of poly(I:C) for the indicated time, or transfection of poly(dA:dT) or 5′ppp-dsRNA for 2 h. Densitometry shown in Supplementary Fig. 2c and d. **k** Immunofluorescence of HeLa cells transfected with TRAF3IP3-mCherry or mCherry EV, followed by mock transfection or transfection of poly(I:C) for 3 h. Scale bar, 20 μM. **l** IFN-β ELISA using HEK293T cells transfected with Myc-TRAF3IP3 or EV, followed by mock transfection, transfection of poly(I:C), poly(dA:dT), 5′ppp-dsRNA for 9 h or VSV infection (MOI = 0.5) for 9 h. Data are presented as mean ± SEM and are one representative of three independent experiments. **a–h** One-way ANOVA followed by Dunnet post hoc correction. **l** *t*-test. *$p < 0.05$, **$p < 0.01$, ***$p < 0.001$. Source data are provided as a Source Data file.

TRAF3IP3. Profiling of TRAF3IP3 expression in human cell lines using the Human Protein Atlas revealed the enriched TRAF3IP3 expression in T and B cell lines as well as several myeloid cell lines such as THP-1 and HL60 cells, but not in other commonly used stromal cell lines such as HEK293 or HeLa cells and most other cell lines (Supplementary Fig. 3a). We confirmed the expression of TRAF3IP3 at both mRNA and protein levels in THP-1 cells, and the lack of expression in HEK293T or HeLa cells (Fig. 2a). The endogenous TRAF3IP3 expression in THP-1 cells was similar to that of Jurkat T cells, paralleling its expression in primary T and myeloid cells. HEK293T overexpressing TRAF3IP3 was used as a positive control for antibody specificity, while HEK293T cells served as a negative control.

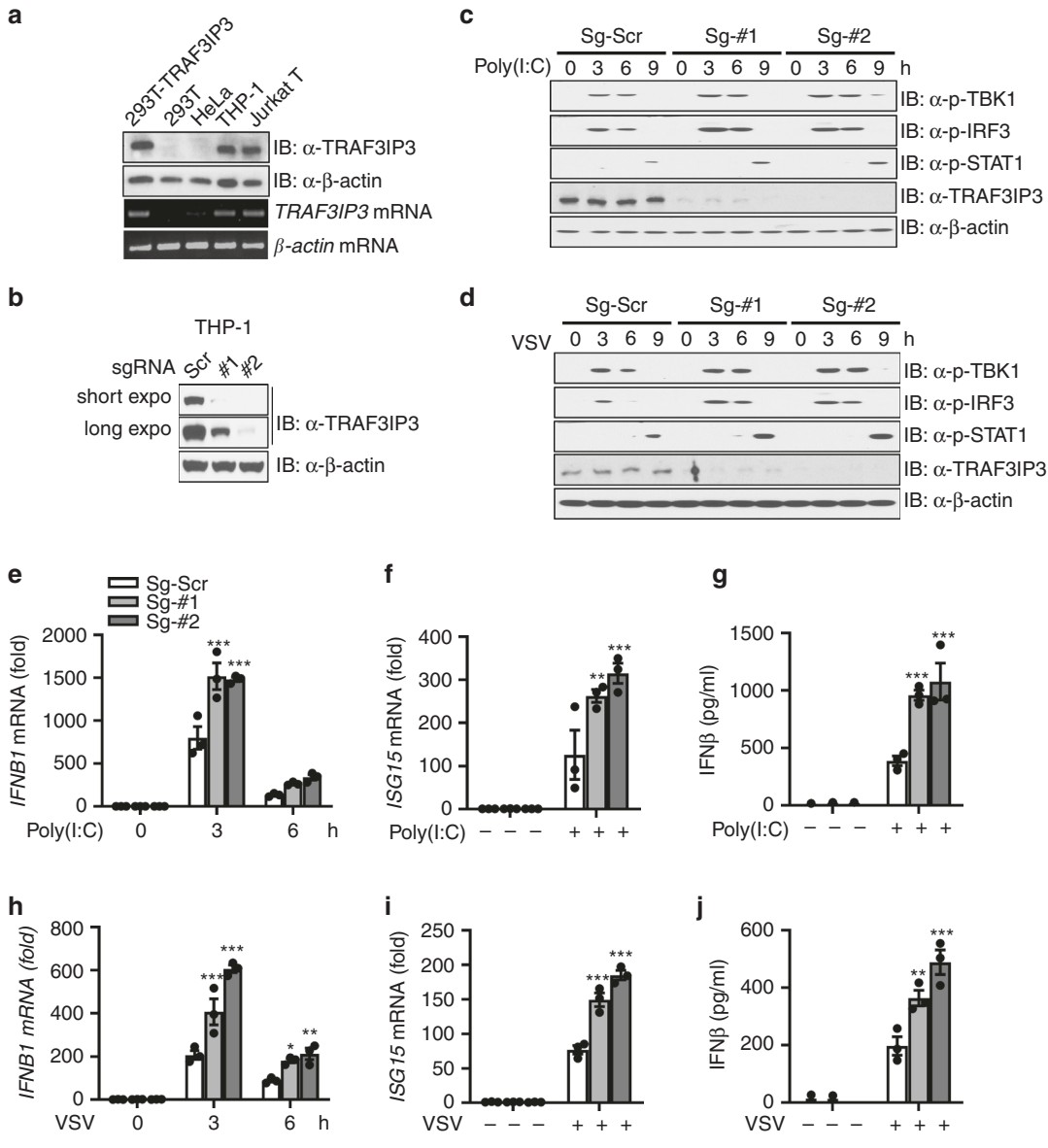

**Fig. 2 CRISPR/Cas9 knockout of TRAF3IP3 potentiates the type I interferon response. a** Immunoblotting (top) and RT-PCR (bottom) analyses using indicated cell lines. HEK293T cells transfected with Myc-TRAF3IP3 serves as the positive control. **b** Immunoblotting of THP-1 cells with scramble sgRNA or sgRNA targeting *TRAF3IP3* using CRISPR/Cas9. **c, d** Immunoblotting of THP-1 cells with scramble sgRNA or sgRNA targeting *TRAF3IP3* transfected with poly(I:C) (**c**) or infected with VSV (MOI = 1) (**d**) for the indicated time. Densitometry of immunoblots in this Figure is shown in Fig. S4. **e, f** RT-PCR analysis and **g** IFN-β ELISA using THP-1 cells with scramble sgRNA or sgRNA targeting *TRAF3IP3* transfected with poly(I:C). RT-PCR data were normalized to *ACTB* mRNA. **h, i** RT-PCR analysis and **j** IFN-β ELISA using THP-1 cells with scramble sgRNA or sgRNA targeting *TRAF3IP3* infected with VSV (MOI = 1). RT-PCR data were normalized to *ACTB* mRNA. **f, g, i** and **j** were at 9 h post transfection or infection. sgRNA, small guide RNA. Scr, Scramble. Data are presented as mean ± SEM and are one representative of three independent experiments. **e–j**, two-way ANOVA followed by Holm-Sidak post hoc correction. *$p < 0.05$, **$p < 0.01$, ***$p < 0.001$. Source data are provided as a Source Data file.

We opted to use THP-1 cells in further studies because it is an innate immune cell line, specifically a myeloid-monocytic cell line, and displays an intact IFN-I response to viral RNA. To confirm the role of TRAF3IP3 in modulating cytosolic RNA-induced IFN-I signaling, we transduced THP-1 cells with CRISPR/Cas9 lentivirus expressing a scramble (Scr) small guide RNA (sgRNA) or two different sgRNAs targeting *TRAF3IP3*. These two sgRNAs resulted in the efficient reduction of *TRAF3IP3* in THP-1 cells (Fig. 2b, and Supplementary Fig. 4a). Cytosolic poly(I:C) stimulation and VSV infection are known to cause phosphorylation of TBK1, which phosphorylates and activates IRF3. The phosphorylation of TBK1 and IRF3 was enhanced in the THP-1 cells with reduced TRAF3IP3 expression

relative to control cells after cytosolic poly(I:C) stimulation or VSV infection (Fig. 2c, d, densitometric measurements shown in Supplementary Fig. 4b, c, e, f). We also found enhanced STAT1 phosphorylation (Fig. 2c, d, densitometric measurements shown in Supplementary Fig. 4d, g), which is known to be activated by IFN-I. THP-1 cells with reduced TRAF3IP3 expression also produced more *IFNB1* mRNA, IFN-β protein and interferon-stimulated gene 15 (*ISG15*) mRNA than control cells (Fig. 2e–j). However, *TRAF3IP3* mRNA was stable in THP-1 cells after Poly(I:C) transfection or VSV infection (Supplementary Fig. 4h, i). Similarly, profiling of *Traf3ip3* in RAW264.7 cells using Infectome Map[38] showed *Traf3ip3* also appeared stable after VSV challenge (Supplementary Fig. 4j). Thus, *TRAF3IP3* is not

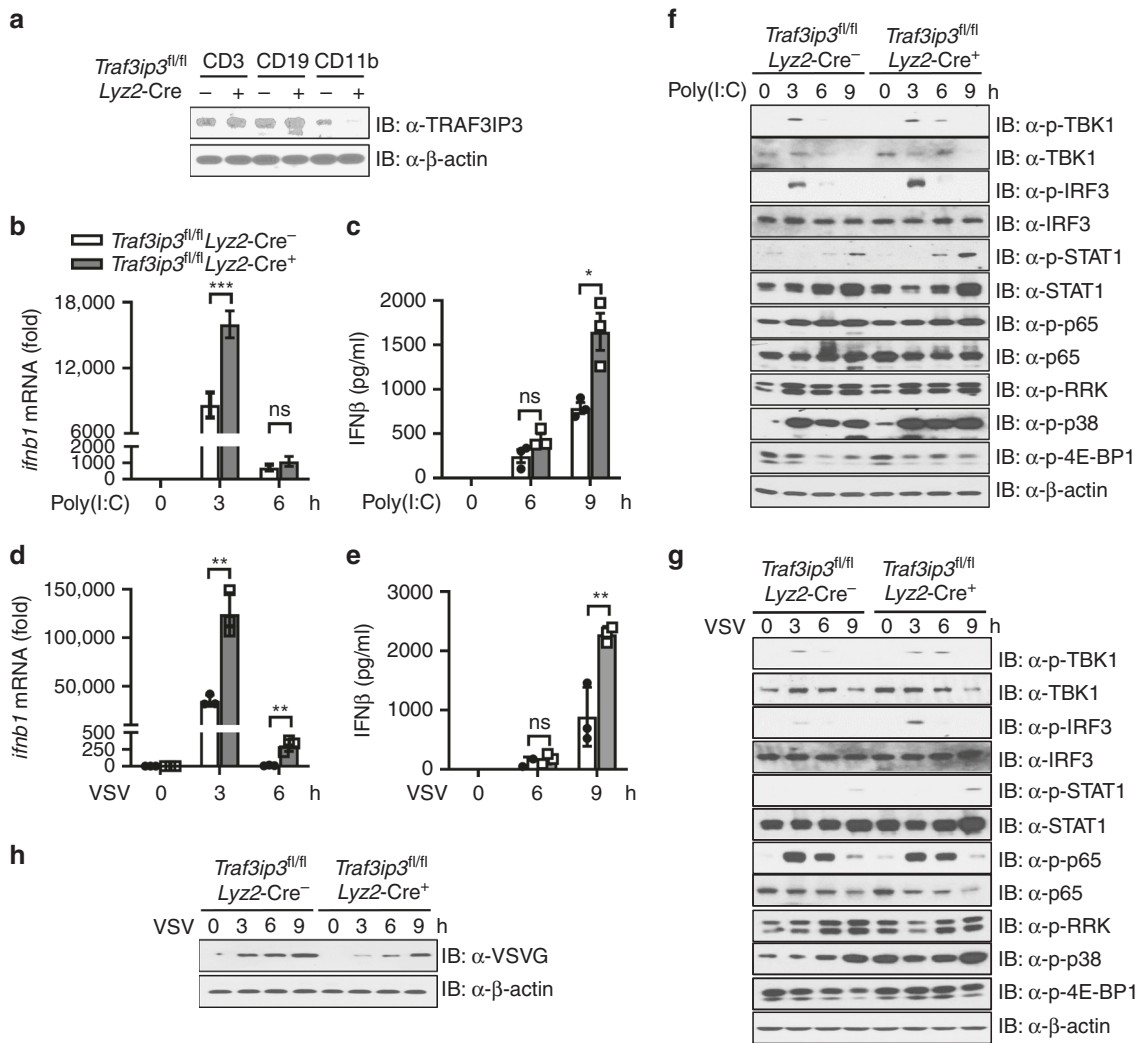

**Fig. 3 *Traf3ip3* deficiency potentiates the type I interferon response. a** Immunoblotting of flow sorted CD3[+] T cells, CD19[+] B cells and CD11b[+] myeloid cells from indicated mice. **b** RT-PCR analysis and **c** IFN-β ELISA using *Traf3ip3*[fl/fl] *Lyz2*-Cre[−] and *Traf3ip3*[fl/fl] *Lyz2*-Cre[+] BMDMs transfected with poly(I:C) for the indicated times. RT-PCR data were normalized to *Actb* mRNA. **d** RT-PCR analysis and **e** IFN-β ELISA using *Traf3ip3*[fl/fl] *Lyz2*-Cre[-] and *Traf3ip3*[fl/fl] *Lyz2*-Cre[+] BMDMs infected with VSV (MOI = 1) for the indicated times. RT-PCR data were normalized to *Actb* mRNA. **f, g** Immunoblotting of signaling proteins using *Traf3ip3*[fl/fl] *Lyz2*-Cre[−] and *Traf3ip3*[fl/fl] *Lyz2*-Cre[+] BMDMs transfected with poly(I:C) (**f**) or infected with VSV (MOI = 1) (**g**) for the indicated time. Densitometry shown in Supplementary Fig. 6a and b. **h** Immunoblotting of VSVG using *Traf3ip3*[fl/fl] *Lyz2*-Cre[-] and *Traf3ip3*[fl/fl] *Lyz2*-Cre[+] BMDMs infected with VSV (MOI = 2) for the indicated time. Densitometry shown in Supplementary Fig. 6c. Data are presented as mean ± SEM and are one representative of three independent experiments. **b–e** n = 3 biologically independent animals, t-test. *p < 0.05, **p < 0.01, ***p < 0.001. ns, not significant. Source data are provided as a Source Data file.

transcriptionally targeted by IRF3 and NF-κB activated by RNA sensing. Together these data suggest reduced TRAF3IP3 potentiates cytosolic RNA-induced IFN-I signaling.

***Traf3ip3* deficiency potentiates the type I interferon response.** TRAF3IP3 was shown to be essential for T and B cell development[32–34]. To specifically investigate the role of TRAF3IP3 in innate immunity, we generated *Traf3ip3*[fl/fl]*Lyz2*-Cre[+] mice, which harbored deletion of the loxP-flanked *Traf3ip3* (*Traf3ip3*[fl/fl]) specifically in myeloid cells via Cre recombinase under control of the myeloid-cell specific promoter Lyz2 (*Lyz2*-Cre). *Traf3ip3*[fl/fl]*Lyz2*-Cre[+] mice appeared to have normal development of myeloid cells in the bone marrow, lymph node, and peritoneal lavage (Supplementary Fig. 5a–d). As controls, T and B cell development was normal in mice lacking *Traf3ip3* only in the myeloid component, as expected (Supplementary Fig. 5a–d). To confirm that TRAF3IP3 was specifically deleted in myeloid cells, we sorted the CD3[+] T cells,

CD19[+] B cells and CD11b[+] myeloid cells by flow cytometry and confirmed that TRAF3IP3 was deleted in the CD11b[+] myeloid cells but not CD3[+] T cells or CD19[+] B cells (Fig. 3a). To investigate the function of TRAF3IP3 in the cytosolic RNA sensing pathway, we generated bone marrow derived macrophages (BMDMs) from control *Traf3ip3*[fl/fl]*Lyz2*-Cre[−]and targeted *Traf3ip3*[fl/fl]*Lyz2*-Cre[+] mice. Cytosolic poly(I:C) stimulation (Fig. 3b, c) or VSV infection (Fig. 3d, e) induced more *Ifnb1* mRNA and IFN-β protein in *Traf3ip3*[fl/fl] *Lyz2*-Cre[+] BMDMs than in *Traf3ip3*[fl/fl]*Lyz2*-Cre[−] BMDMs. In line with this finding, the deficiency of *Traf3ip3* also caused enhanced phosphorylation of TBK1, IRF3, STAT1 and p38, whereas phosphorylation of p65 and ERK appeared comparable between *Traf3ip3*[fl/fl]*Lyz2*-Cre[−] BMDMs and *Traf3ip3*[fl/fl]*Lyz2*-Cre[+] BMDMs (Fig. 3f, g, densitometric measurements shown in Supplementary Fig. 6a, b). TRAF3IP3 was found to inhibit MTOR signaling in Treg cells[32]. The data with 4E-BP1 was varied in that we observed no difference in the phosphorylation of 4E-BP1

after poly(I:C) transfection and a modest increase its phosphorylation in $Traf3ip3^{fl/fl}$ $Lyz2$-Cre$^+$ BMDMs than in $Traf3ip3^{fl/fl}Lyz2$-Cre$^-$ BMDMs (Fig. 3f, g, densitometric measurements shown in Supplementary Fig. 6a, b). $Traf3ip3^{fl/fl}Lyz2$-Cre$^+$ BMDMs also had reduced levels of the VSV specific protein VSV-G than in $Traf3ip3^{fl/fl}Lyz2$-Cre$^-$ BMDMs (Fig. 3h). These data suggest that TRAF3IP3 is a negative regulator of the antiviral immune response to cytosolic RNA and viral infection.

**TRAF3IP3 inhibits RIG-I-MAVS-TBK1 signaling**. Cytosolic RNA is sensed by RIG-I-like receptors which then activate MAVS. MAVS recruits TBK1 to phosphorylate and activate IRF3. Additionally, IRF3 can be activated by TLR3 or TLR4, which surveys extracellular RNA and LPS. In this case, TLR stimulation activates TRIF, leading to the recruitment of TBK1 and subsequent IRF3 activation[5,6]. Overexpression of these components is known to activate the ISRE luciferase reporter. To dissect the role of TRAF3IP3 in these different IFN-I induction pathways, we examined the effect of TRAF3IP3 on the ISRE luciferase reporter activated by these components. Overexpression of TRAF3IP3 reduced the ISRE activation by RIG-I and to a greater extent by TBK1, but its effect on MAVS is moderate (Fig. 4a–c). However, TRAF3IP3 did not inhibit downstream IRF3 dependent ISRE activation (Fig. 4d), indicating that TRAF3IP3 targets the pathway at nodes upstream of IRF3. TRAF3IP3 also inhibited TRIF (Fig. 4e). In addition to the overexpressed TRAF3IP3, silencing of TRAF3IP3 in TRAF3IP3 overexpressing cells by siRNA knockdown (Fig. 4f) significantly mitigated the TRAF3IP3-mediated suppression of TBK1 dependent ISRE activation (Fig. 4g). Overexpression of TRAF3IP3 also suppressed endogenous IRF3 phosphorylation induced by overexpression of TBK1 (Fig. 4h), with moderate effects on RIG-I (Fig. 4i) and TRIF (Fig. 4j), and a slight impact on MAVS (Fig. 4k) but not on IRF3 (Fig. 4l). Therefore, TRAF3IP3 suppresses the IFN-I signaling at a step upstream of IRF3 with the most impact on TBK1.

**TRAF3IP3 targets TRAF3 and TBK1**. To explore the molecular mechanism of how TRAF3IP3 interferes with the IFN-I pathway, we tested whether TRAF3IP3 can interact with the components in this pathway. In an overexpression system, coimmunoprecipitation followed by immunoblotting revealed TRAF3IP3 interacted strongly with TBK1 (Fig. 5a). TRAF3IP3 did not interact with RIG-I, MAVS, IRF3 (Fig. 5a). Next, reciprocal domain mapping experiments were conducted with TBK1 deletion constructs and TRAF3IP3 deletion constructs. The N-terminus of TBK1, which contained the kinase domain, was required for TRAF3IP3 association (Fig. 5b). For TRAF3IP3, the domain that showed the strongest interaction with TBK1 is residues 455–551 followed by residues 1–235 (Fig. 5c).

TRAF3 was reported to be essential in the TLR-dependent IFN-I response to RNA virus and the TLR-independent IFN-I response to cytosolic RNA[39,40]. MAVS and TRIF each interacts with TRAF3, forming a scaffold to recruit TBK1[39–41]. However, TRAF3 is not required for STING to activate TBK1[42]. TRAF3IP3 was originally reported as a TRAF3 interacting protein[31]. Indeed, we found TRAF3IP3 interacted with TRAF3 but not with TRAF1 (Fig. 5d).

The above experiments relied on overexpression systems, hence it was important to determine if TRAF3IP3 interacts with TBK1 and TRAF3 at the endogenous level. We infected THP1 cells with VSV at various time point and immunoprecipitated TRAF3IP3 associated proteins. The interactions of TBK1 and TRAF3 to TRAF3IP were verified with endogenous proteins and these interactions increased after VSV infection (Fig. 5e). In addition, we co-expressed

MAVS and TBK1, or MAVS and TRAF3, together with increasing amounts of TRAF3IP3. After pulling down TBK1 or TRAF3, the amount of MAVS bound to TBK1 or TRAF3 was reduced in the presence of increasing amounts of transfected TRAF3IP3 (Fig. 5f, g), indicating that TRAF3IP3 competes with MAVS to associate with TRAF3 and TBK1.

**TRAF3IP3 promotes TBK1 ubiquitination and degradation**. To test whether additional mechanisms are used by TRAF3IP3 to target this pathway, we co-overexpressed TBK1 and IRF3 with an increasing amount of TRAF3IP3. TBK1 protein expression diminished with increasing TRAF3IP3 expression levels (Fig. 6a, also see Fig. 5f, input lane). Consistent with decreased TBK1 protein, IRF3 phosphorylation was also reduced with increasing TRAF3IP3 expression (Fig. 6a). However, IRF3 protein expression was not altered by increasing TRAF3IP3 expression. The reduced TBK1 protein could be due to reduced $TBK1$ gene expression but RT-PCR showed $TBK1$ mRNA was not altered by increasing TRAF3IP3 expression (Fig. 6a). Therefore, the main detectable effect is the reduction of TBK1 protein in the presence of TRAF3IP3. Conversely, increased TBK1 protein was also observed earlier in primary cells from $Traf3ip3^{fl/fl}Lyz2$-Cre$^+$ macrophages compared to control macrophages (Fig. 3f, g).

Ubiquitination of TBK1 is important for its activation and degradation. K63-linked polyubiquitination promotes TBK1 activation, whereas K48-linked polyubiquitination promotes its degradation[24,26,29]. To test if TRAF3IP3 affected TBK1 ubiquitination, we co-overexpressed TBK1 with or without TRAF3IP3, in the presence of a ubiquitin expressing vector. Cells were lysed in a stringent 1% SDS buffer and denatured by boiling so that only covalent modifications remained, followed by immunoprecipitation of TBK1. TBK1 polyubiquitination was increased in the presence of TRAF3IP3 (Fig. 6b). We then used introduced ubiquitin K48 or K63 mutants where all Lys except K48 or K63 were mutated to Arg. TRAF3IP3 specifically increased K48-linked but not K63-linked polyubiquitination of TBK1 (Fig. 6c), suggesting that TRAF3IP3 increased the degradative ubiquitination of TBK1. These changes in TBK1 levels were also seen in the anti-Flag lanes which detected TBK1. K63-linked polyubiquitination of TBK1 was reduced by TRAF3IP3 but this is likely due to reduced input TBK1.

To corroborate the impact of TRAF3IP3 on K48 ubiquitination of TBK1 endogenously in primary cells, we immunoprecipitated endogenous TBK1 from $Traf3ip3^{fl/fl}Lyz2$-Cre$^-$ or $Traf3ip3^{fl/fl}Lyz2$-Cre$^+$ BMDMs infected with VSV. TBK1 from $Traf3ip3^{fl/fl}Lyz2$-Cre$^-$BMDMs had more K48-linked polyubiquitination than from $Traf3ip3^{fl/fl}Lyz2$-Cre$^+$ BMDMs (Supplementary Fig. 7a). Ubiquitination requires the sequential actions of three enzymes: Ub-activating enzyme (E1), Ub-conjugating enzyme (E2), and Ub ligase (E3), and the E3 ligase dictates which target protein gets ubiquitinated[26,28,29]. Several E3 ubiquitin-protein ligases including DTX4, TRIP and TRIM27 have been reported to induce TBK1 K48-linked polyubiquitination[26,28,29]. We reduced the expression of DTX4, TRIP or TRIM2 using shRNA (Supplementary Fig. 7b) and found that the reduction of DTX4 decreased the TRAF3IP3 mediated K48-linked polyubiquitination of TBK1 (Fig. 6d). Furthermore, we tested multiple TBK1 constructs bearing Lys to Arg mutations in possible ubiquitination sites, and identified Lys372 in the ULD domain of TBK1 as the critical target for TRAF3IP3 mediated K48-linked polyubiquitination of TBK1 (Fig. 6e). Since tight regulation of TBK1 is critical for IFN-I signaling, these results indicate that TRAF3IP3 might inhibit IFN-I signaling by promoting K48-linked polyubiquitination and degradation of TBK1.

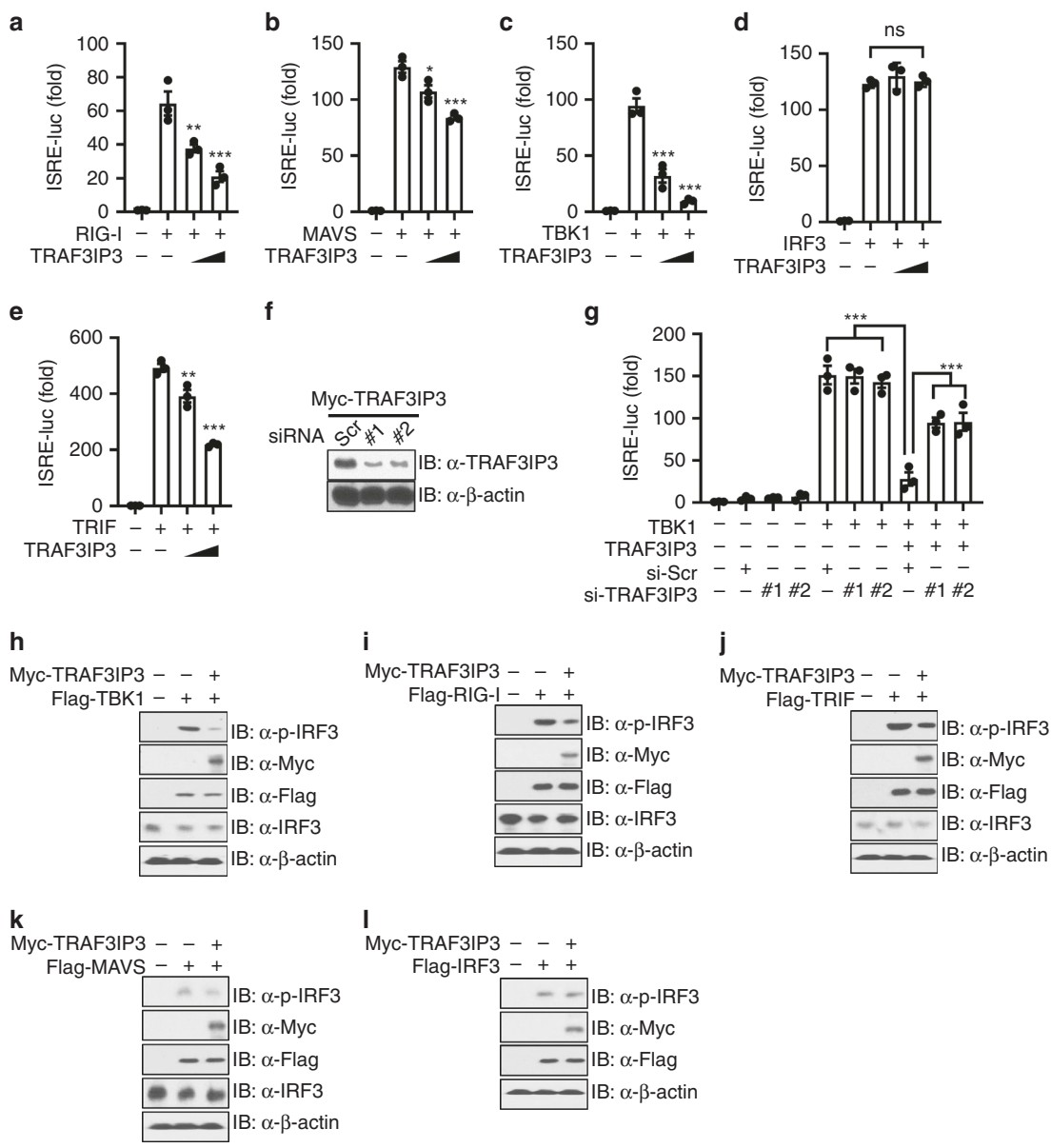

**Fig. 4 TRAF3IP3 inhibits RIG-I-MAVS-TBK1 signaling. a–e** Luciferase assay in HEK293T cells transfected with ISRE reporter, Flag-tagged indicated expression vectors, along with empty vector or increasing doses of Myc-TRAF3IP3 vector (wedge). *Renilla* luciferase was used as the internal control. **f** Immunoblotting of HEK293T cells transfected with Myc-TRAF3IP3 together with scramble and two *TRAF3IP3* targeting siRNA. **g** Luciferase assay in HEK293T cells transfected with ISRE reporter, EV, HA-TBK1 or Myc-TRAF3IP3 vector, together with scramble or two *TRAF3IP3* targeting siRNA. *Renilla* luciferase was used as the internal control. **h–l** Immunoblotting of HEK293T cells transfected with Myc-TRAF3IP3 and the indicated FLAG-tagged expression vectors. Data are presented as mean ± SEM and are one representative of three independent experiments. **a–e, g**, one-way ANOVA followed by Dunnet post hoc correction. *$p < 0.05$, **$p < 0.01$, ***$p < 0.001$. ns, not significant. Source data are provided as a Source Data file.

***Traf3ip3* deficient mice are more resistant to VSV infection**. To assess the physiologic relevance of this study, we evaluated the importance of TRAF3IP3 in antiviral host defense in vivo. *Traf3ip3*$^{fl/fl}$*Lyz2*-Cre$^+$ and *Traf3ip3*$^{fl/fl}$*Lyz2*-Cre$^-$ mice were challenged with VSV intraperitoneally and monitored for survival, weight change, and morbidity. Both groups of mice rapidly lost body weight in the first 5 days post-infection. However, *Traf3ip3*$^{fl/fl}$*Lyz2*-Cre$^+$ mice had less body weight loss by the third-day post-infection and had significantly more body weight recovery after 10 days (Fig. 7a). All infected *Traf3ip3*$^{fl/fl}$*Lyz2*-Cre$^-$ mice died by day 14, whereas 40% *Traf3ip3*$^{fl/fl}$*Lyz2*-Cre$^+$ mice remained alive at day 14 post-infection (Fig. 7b). *Traf3ip3*$^{fl/fl}$*Lyz2*-Cre$^+$ mice had increased serum IFN-β at 8 h post-infection

(Fig. 7c), as well as increased *Ifnb1* mRNA expression in lung, liver and spleen 24-h post-infection (Fig. 7d). *Traf3ip3*$^{fl/fl}$*Lyz2*-Cre$^+$ mice also had reduced VSV-G and VSV viral titers in the lung, liver and spleen at 24-h post-infection (Fig. 7e–f, densitometric measurements shown in Supplementary Fig. 6c). Together, these data suggest myeloid *Traf3ip3* deficiency enhances viral clearance and diminishes morbidity caused by VSV infection.

## Discussion
The elucidation of immune regulatory mechanisms is crucial to understanding how the host constrains adventitious inflammation and maintains immune homeostasis. In the present study, we demonstrate an essential role of TRAF3IP3 in the negative

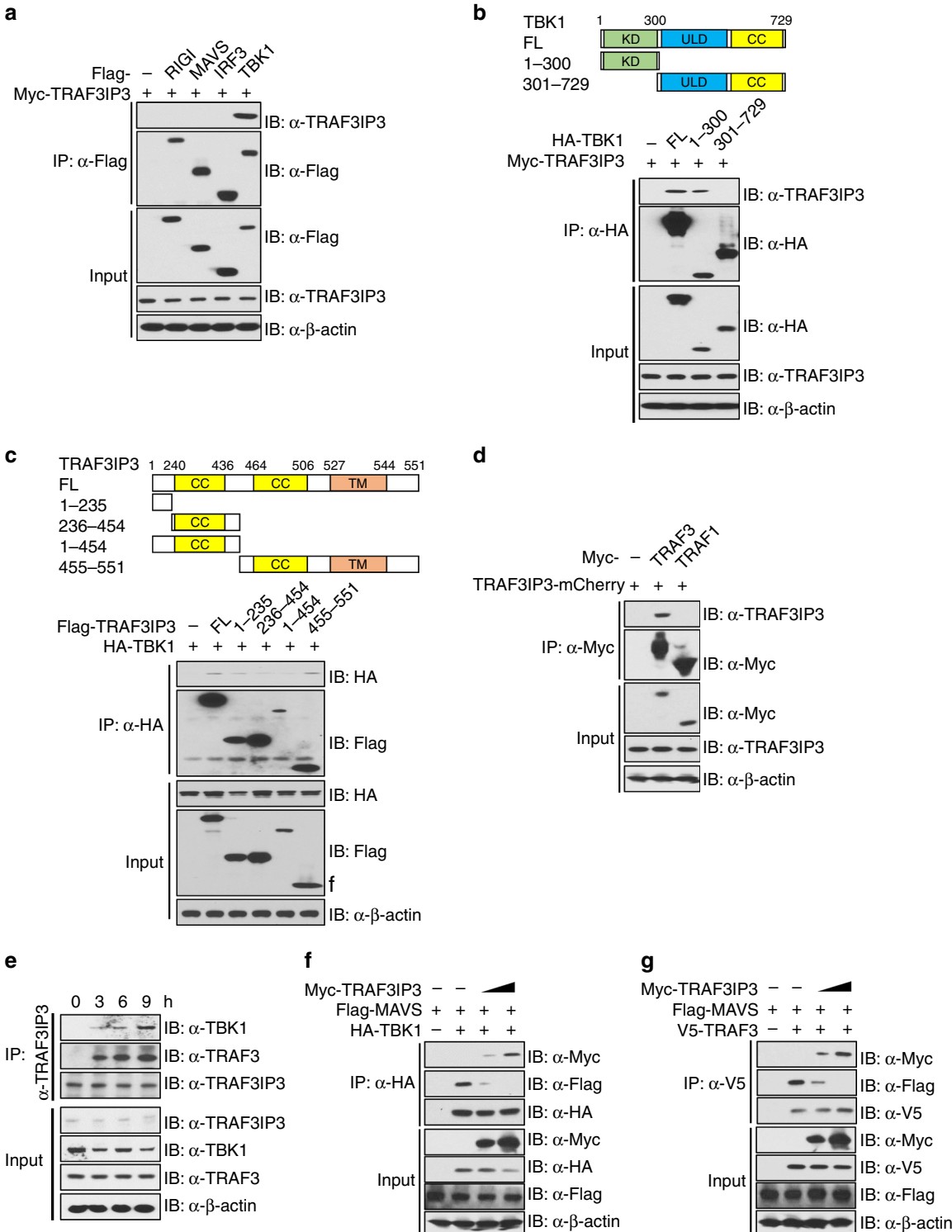

**Fig. 5 TRAF3IP3 targets TRAF3 and TBK1. a–d** Coimmunoprecipitation and immunoblotting using lysates from HEK293T cells transfected with the indicated expression vectors. **e** Immunoblotting of endogenous proteins with the indicated antibodies using lysates of THP-1 cells infected for various times with VSV (MOI = 1) followed by immunoprecipitation with anti-TRAF3IP3. **f, g** Coimmunoprecipitation and immunoblotting using lysates from HEK293T cells transfected with the indicated expression vectors. Data are one representative of at least three independent experiments. Source data are provided as a Source Data file.

regulation of an innate anti-viral pathway in myeloid cells (Fig. 8). TRAF3IP3 reduced the IFN-I response by cytosolic RNA, which provided a clue linking TRAF3IP3 to the RLR pathway. Deficiency of TRAF3IP3 in both human and mouse cells potentiated IFN-I induction by cytosolic RNA. TRAF3IP3 associated

with both TRAF3 and TBK1 and prevented downstream signaling transduction. Additionally, TRAF3IP3 promoted K43-linked polyubiquitination and degradation of TBK1. Importantly, ablation of *Traf3ip3* led to enhanced anti-viral cytokine production and viral clearance in vivo, underscoring its physiologic function.

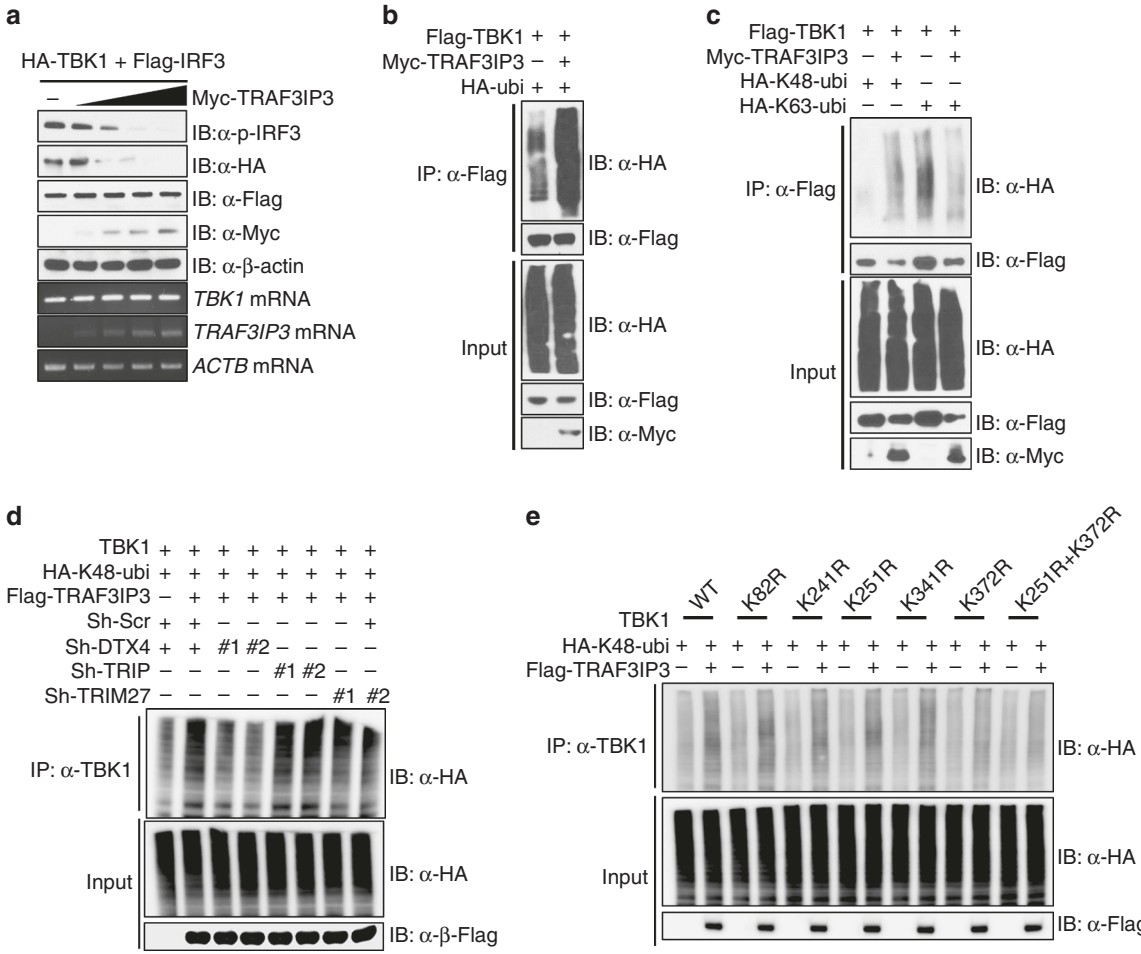

**Fig. 6 TRAF3IP3 promotes K48-dependent ubiquitination of TBK1. a** Immunoblotting (top) and RT-PCR (bottom) analyses using HEK293T transfected with HA-TBK1 and Flag-IRF3 and increasing doses of Myc-TRAF3IP3. **b-e** Coimmunoprecipitation and immunoblotting using HEK293T transfected with indicated vectors. Data are one representative of at least two independent experiments. Source data are provided as a Source Data file.

This work demonstrates for the first time that TRAF3IP3 serves as an immunological rheostat to safeguard against inappropriate inflammatory responses to cytosolic viral RNA in myeloid cells.

TRAF3IP3 was originally identified as a TRAF3 interacting protein[31]. Later studies showed that TRAF3IP3 is required for the maturation of double positive T cell to mature T cell by facilitating ERK signaling[33]. In addition, a more recent study showed that TRAF3IP3 is essential for maintaining regulatory T cell stability and function by suppressing the mechanistic target of rapamycin complex 1 (mTORC1)[32]. Cytosolic RNA stimulation is known to activate ERK signaling but inhibit mTORC1 signaling[43,44], which is also consistent with data shown in wildtype controls. However, we did not observe a difference in p-ERK signaling in *Traf3ip3* deficient BMDMs and the difference of p-4EBP1 appeared inconsistent after Poly(I:C) and VSV stimulation. Instead, we consistently observed enhanced p-TBK1 and p-IRF3 signaling in *Traf3ip3*-deficient BMDMs. In addition, we observed increased p-STAT1 and p-p38, both of which are activated by IFN-I and hence downstream of IFN-I signaling to IFN receptor[45,46]. Thus, TRAF3IP3 appears to have multiple functions, depending on the cell type. Our data reveal a role of TRAF3IP3 in modulating the RLR sensing pathway in innate immunity, although its impact is likely to be extended to other PRR pathways since this report also shows that it reduced TRIF activation.

Upon sensing cytosolic RNA, RIG-I and MDA5 activate MAVS and induce MAVS to form prion-like aggregates[16]. Crucial to MAVS complex formation and downstream signaling is the recruitment of TRAFs, including TRAF2, TRAF3, TRAF5, and TRAF6[16,40,41]. Interaction of MAVS with TRAF2 or TRAF6 is involved in IKK-dependent NK-κB activation, whereas TRAF3 is specifically involved in TBK1-dependent IRF3 activation[40,41]. TRAF3-mediated K63-linked poly-ubiquitination of its substrate and of TRAF3 itself serves as a scaffold for the recruitment of TBK1-IRF3, albeit TBK1 is not a direct substrate of TRAF3. Our data do not suggest a role of TRAF3IP3 in modulating NK-κB activation downstream of RLR signaling as p-p65 signaling was comparable between control and *Traf3ip3* deficient BMDMs. We found TRAF3IP3 associated with TRAF3 and competed with MAVS for binding TRAF3, thus showcasing a putative molecular mechanism for how TRAF3IP3 specifically attenuates the TBK1-IRF3 axis. Post-translational modification is also important for TBK1 activation and degradation. Additionally, we also found that TRAF3IP3 promotes K48-linked polyubiquitination and degradation of TBK1, providing another layer of molecular intricacy of how TRAF3IP3 modulates the RLR pathway. We also show that this process is dependent on the E3 ligase, DTX4. The TRAF3-TBK1 complex is also central to the TLR-dependent IFN-I response[39]. Downstream of TLRs, TRIF is an essential adaptor protein recruiting TRAF3-TBK1. We also found that TRAF3IP3 potently inhibited the TRIF-dependent ISRE reporter. Thus TRAF3IP3 may also be involved in regulating the TLR3 response.

In summary, our findings show the attenuation of cytosolic viral sensing by TRAF3IP3 and reveal that TRAF3IP3 represents an innate immune checkpoint for TBK1-induced inflammation.

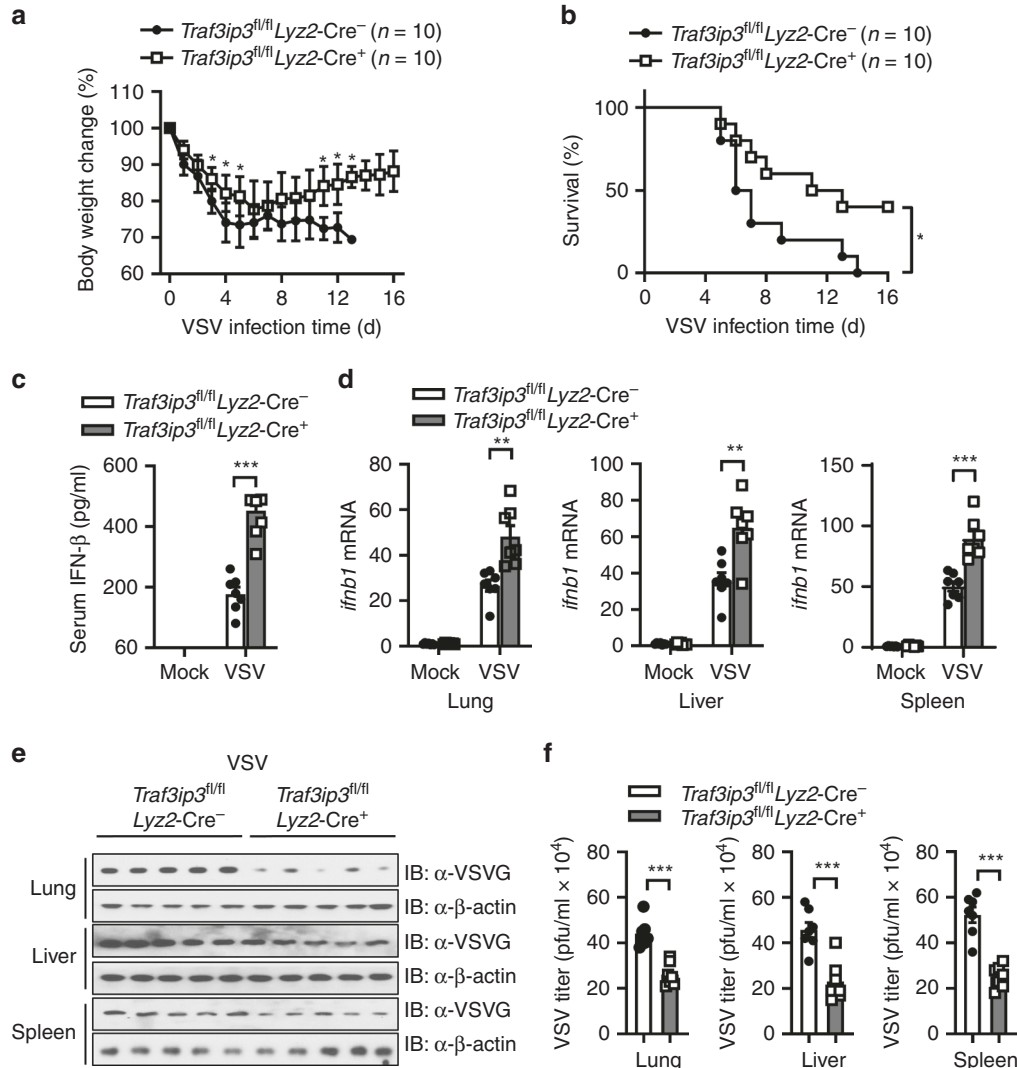

**Fig. 7 Traf3ip3 deficient mice are resistant to VSV infection. a**, **b** Body weight change and survival rate of Traf3ip3[fl/fl] Lyz2-Cre⁻ and Traf3ip3[fl/fl]Lyz2-Cre⁺ mice infected with VSV ($0.5 \times 10^7$ pfu per mouse) by i.p. injection. **c** ELISA of serum IFN-β from Traf3ip3[fl/fl] Lyz2-Cre⁻ and Traf3ip3[fl/fl] Lyz2-Cre⁺ mice at 8-h post VSV infection or mock infection. **d** RT-PCR of Ifnb1 mRNA in the lung (left), liver (middle) and spleen (right) from infected mice at 24-h post VSV infection or mock infection. RT-PCR data were normalized to Actb mRNA. **e** Immunoblotting of VSV-G in the lung, liver and spleen from infected mice as in (**d**). Each lane represents a sample from an individual mouse. Densitometry shown in Supplementary Fig. 6c. **f** Plaque assay of VSV in the lung, liver, spleen from infected mice as in (**d**). Data are presented as mean ± SEM. Data in a and b are pooled from three independent experiments, $n = 10$ biologically independent animals in total. Data in (**c**), (**d**) and (**f**) are pooled from two independent experiments, $n = 7$ biologically independent animals in total. **a** t-test. **b** Log-rank (Mantel-Cox) test. **c**, **d** and **f** t-test. *$p < 0.05$, **$p < 0.01$, ***$p < 0.001$. Source data are provided as a Source Data file.

This work expands the regulatory landscape of the cytosolic sensing by RNA receptors such as the RLR sensing pathway and uncovers a function of TRAF3IP3 in modulating innate immunity that was not previously appreciated.

## Methods

**Animal husbandry**. All animal protocols were approved by the Institutional Animal Care and Use Committee (IACUC, protocol 19-123) of University of North Carolina Chapel Hill (UNC) in accordance with the National Institutes of Health Guide for the Care and Use of Laboratory Animals. All animal experiments were performed under specific pathogen–free conditions in sterile isolated cages using 8-week-old mice. Mice were kept at maximum of 5 mice/cage with corn cob bedding at ambient temperature of 20–22 °C, 40–60% humidity under 7 am to 7 pm light cycle, 7 pm to 7 am dark cycle. Mice were breed on the C57BL/6 background and age and sex matched mice were used. Traf3ip3[fl/fl] mice were previously reported[33] and crossed to Lyz2-Cre mice (The Jackson Laboratory, stock 4781) to generate myeloid cell deletion targeted Traf3ip3[fl/fl]Lyz2-Cre⁺ and control Traf3ip3[fl/fl]Lyz2-Cre⁻ mice. No randomization of the allocation of animals to experimental groups was performed.

**Cell lines and virus**. HEK293T, HeLa, Vero, THP-1, Jurkat T, BJAB cells were acquired from the Tissue Culture Core Facility from UNC at Chapel Hill. HEK293T, Vero and HeLa cells were maintained in complete DMEM (Gibco) supplemented with 10% fetal bovine serum (FBS), 100 U/ml penicillin and 100 µg/ml streptomycin THP-1 and Jurkat T cells were maintained in RPMI-1640 (Gibco) supplemented with 10% FBS, 100 U/ml penicillin and 100 µg/ml streptomycin. The identities of these cells were not authenticated during the course of this study. BMDMs were generated by flushing the mice femurs with PBS and removing red blood cells by ammonium-chloride-potassium (ACK) buffer lysis. Then, BMDMs were differentiated in the presence of L-929 conditional medium (50% DMEM, 20% FBS, 30% L-929 medium, 100 U/ml penicillin and 100 µg/ml streptomycin) for 5 days. All cells were grown in a 37 °C incubator supplied with 5% CO2. VSV virus was propagated in Vero cells. Viral titers were determined using a plaque assay with Vero cells.

**Cell stimulation**. BMDMs were seeded into 24-well plates at $1 \times 10^6$ per well in complete DMEM medium and transfected with 1 µg poly (I:C) (Invivogen, #tlrl-pic), 1 µg poly(dA:dT) (Invivogen, #ttlrl-patn), or 1 µg 5'ppp-RNA (Invivogen, #tlrl-3prna) by lipofectamine 2000 (Thermo Scientific, #11668027) for the indicated time. BMDMs were infected with VSV at the indicated MOIs at 37 °C for 1 h. Cells were then washed with warm PBS and cultured in complete DMEM. THP-1

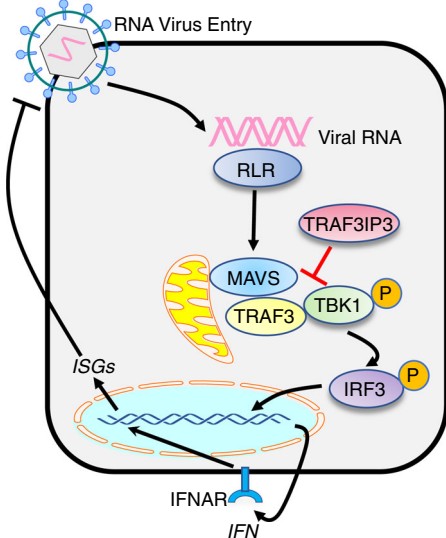

**Fig. 8 Schematic illustration of TRAF3IP3 serving as immune rheostat by inhibiting RLR dependent IFN-I pathway.** TRAF3IP3 inhibits the RLR sensing pathway by targeting at node of TBK1 and TRAF3.

were seeded into 24-well plates at $0.5 \times 10^6$ per well in complete RPMI medium and transfected with 1 μg poly (I:C) using lipofectamine® LTX (Thermo Scientific, #15338030) for the indicated time.

**Plasmid construction.** Myc-TRAF3IP3, Myc-TRAF3 and Myc-TRAF1 were cloned into pcDNA3. TRAF3IP3 was in-frame cloned into the pmCherry-N1 vector (Takara, #632523) to generate TRAF3IP3-mCherry. The Flag-MAVS, Flag-STING, Flag-TBK1, Flag-IRF3, HA-TBK1 FL, 1-300, 301-729, HA-STING vector, pRT-TK Renilla luciferase reporter, ISRE firefly luciferase reporter were previously reported[47,48]. pEF-Bos Flag-RIG-I was kindly provided by Stanley Lemon (UNC-CH) as previously reported[49]. pUNO1-hcGAS-Flag was kindly provided by Blossom A. Damania (UNC-CH) as previously reported[50]. pEF-Bos TRIF-Flag-His were from Addgene (#41550). HA-K48-linked ubiquitin and HA-K63-linked ubiquitin vectors were previously reported[51]. TBK1 WT, K82R, K241R, K251R, K341R, K372R, K251R K372R were previously reported[28]. Flag-TRAF3IP3 FL, 1-235, 236-454, 1-454, 455–551 were cloned into pCMV vector. pLKO.1 GFP shRNA from Addgene (#30323) and used as scramble control. ShRNAs in pLKO.1 targeting TRIM27 (TRCN0000006437, TRCN0000006438), TRIP (TRCN0000033740, TRCN0000033741), DTX4 (TRCN0000236607, TRCN0000236608) were purchased from Sigma. Sg-scramble and Sg-TRAF3IP3-1 and Sg-TRAF3IP3-2 were cloned into the lentiCRISPRv2 vector (Addgene, #52961) following the Addgene protocol. DNA oligos used for SgRNA are listed in Supplementary Table 1.

**Dual luciferase assay.** HEK293T cells were seeded in 24-well plates at a density of $2.5 \times 10^5$ per well and transfected the following day by lipofectamine 2000 (Thermo Fisher Scientific) following the manufacturer's instruction. IFN-β or ISRE firefly luciferase reporter vector (50 ng each) together with 5 ng of pRT-TK Renilla luciferase reporter vector were transfected with increasing doses of the Myc-TRAF3IP3 vector (0, 100, 200 ng) or empty vector. At 24-h post transfection, HEK293T cells were transfected with 1 μg poly (I:C), 1 μg poly(dA:dT), or 1 μg 5'ppp-RNA using lipofectamine 2000, or were infected with VSV at a MOI = 0.5 for 6 h. After 6 h, luciferase activity was measured using the Dual-Luciferase Reporter Assay System (Promega, #E1910). Flag-RIG-I (130 ng), Flag-TBK1 (130 ng), Flag-TRIF (130 ng), Flag-IRF3 (130 ng), Flag-MAVS (40 ng), Flag-cGAS (70 ng) and Flag-STING (70 ng) were transfected with ISRE, pRT-TK Renilla and Myc-TRAF3IP3 vector as described above. Luciferase activity was measured after 24 h.

**siRNA and shRNA knockdown.** HEK293T cells were transfected with Flag-TBK1, Myc-TRAF3IP3 or empty vector along with scramble siRNA (Origene, #SR30004) or, two TRAF3IP3 siRNAs (Origene, #SR312910A, #SR312910C) by lipofectamine 2000. Briefly, 8 pmol siRNA was used per 24 well. SiRNA sequences were listed in Supplementary Table 1. pLKO.1 containing shRNA were transfected with lipofectamine 2000.

**Lentivirus production and transduction.** Lentiviruses were made by co-transfection of 15 μg lentiviral vector, 5 μg pMD2.G (Addgene, #12259), and 10 μg psPAX2 (Addgene, #12260) into 10 cm tissue culture plate bearing 70% confluent HEK293T cells. At 6-h post transfection, the media was replaced with fresh media. At 48 h post transfection, the supernatants containing the lentivirus was collected,

filtered through a 0.45 μm filter, and precipitated using lentivirus precipitation solution (ALSTEM, #VC100) following the manufacturer's instruction. THP-1 cells were transduced by lentiviruses containing sgRNA by spinoculation at $1000 \times g$ at 25 °C for 3 h to generate the TRAF3IP3 knockout cell lines. Transduced cells were selected using 2 μg/ml puromycin at 2 days post lentivirus transduction.

**Indel frequency assay.** The Indel in THP-1 cell generated by CRISPR/Cas9 was determined using Guide-it Mutation Detection Kit at 7 days post lentivirus transduction (Takara, #631448). DNA oligos used for generating *TRAF3IP3* amplicon were listed in Supplementary Table 1. The Indel frequency was determined according to a previous paper[52]. Briefly, integrated intensity of the PCR amplicon and cleaved bands were measured by using Fiji ImageJ. Then the fraction of the PCR product cleaved (fcut) was calculated by using the following formula: $fcut = (b + c)/(a + b + c)$, where $a$ is the integrated intensity of the undigested PCR product and $b$ and $c$ are the integrated intensities of each cleavage product. Then Indel occurrence was estimated with the following formula:

$$\text{Indel}\,(\%) = 100 \times \left(1 - \sqrt{(1 - f\text{cut})}\right) \quad (1)$$

**qRT-PCR.** Total RNA was extracted using TRIzol™ Reagent (Thermo Scientific, #15596026), followed by reverse transcription using iScript™ cDNA Synthesis Kit (BIO-RAD, #1708891) according to manufacturers' protocol. Gene expression was determined using PowerUp™ SYBR™ Green Master Mix (Thermo Scientific, #A25741) on a QuantStudio 6 Flex Real-Time PCR System. Primers used are listed in Supplementary Table 1.

**Confocal microscopy.** HeLa cells were seeded onto coverslips in 24-well dishes. After overnight grown, the cells were transfected with 300 ng T3JAM-mCherry. At 24-h post transfection, 500 ng poly(I:C) were transfected and then after 3 h, cells were fixed with 4% paraformaldehyde (PFA) and permeabilized with PBS with 0.2% Triton X-100. Cells were stained with anti-IRF3 (CST, #11904) (1:200 dilution), followed by AF546-conjugated anti-rabbit secondary antibody (Thermo Scientific, #A-11035) (1:200 dilution) and DAPI. Cells were analyzed using a Zeiss LSM 710 laser-scanning confocal microscope. Images were acquired by ZEN 3.1 software at 63× magnification.

**Enzyme-linked immunosorbent assay.** Human IFN-β was measured using LumiKine™ hIFN-β (Invivogen, #lumi-hifnb) according to the manufacturer's instruction. Mouse IFN-β was measured as follows: Rat monoclonal anti-IFN-β (Santa Cruz, #sc-57201) (1:200 dilution) was used for plate coating; rabbit polyclonal anti-IFN-β (R&D, #32400-1) (1:500 dilution) was used for capturing IFN-β after sample incubation; and goat anti-rabbit IgG HRP (Cell Signaling, #7074) (1:1000 dilution) was used for detection.

**Immunoblotting.** The cells were lysed with RIPA buffer (Boston Bioproducts, BP-116TX) containing protease inhibitor cocktail (Sigma-Aldrich, cOmplete™, #11697498001,) and phosphatase inhibitor cocktail (Sigma-Aldrich, PhosSTOP™, #4906845001) for 30 min on ice. Cell lysates or supernatants were mixed with SDS loading buffer and denatured at 95 °C for 15 min. Then the samples were subjected to 4–12% NuPAGE (Invitrogen) and transferred to nitrocellulose membranes (BIO-RAD, 1620112). The membranes were blocked with 5% milk in TBST buffer for 1 h. The following antibodies were used at 1:2000 dilution: anti-phospho-TBK1 (Ser172, D52C2, CST, #5483), anti-TBK1/NAK (D1B4, CST, #3504), anti-phospho-IRF-3 (Ser396, 4D4G, CST, #4947), anti-IRF-3 (D6I4C, CST, #11904), anti-phospho-Stat1 (Ser727, CST, #9177), anti-Stat1 (CST, #9172), anti-phospho-p65 (Ser536, 93H1, CST, #3033), anti-p65 (D14E12, CST, #8242), anti-phospho-p38 (Thr180/Tyr182, 28B10, CST, #9216), anti-phospho-Erk1/2 (Thr202/Tyr204, D13.14.4E, CST, #4370), anti-phospho-4E-BP1 (Thr37/46, 236B4, CST, #2855), anti-Flag (CST, #14793), anti-HA (CST, #3724), anti-Myc (CST, #2276), anti-V5 (CST, #13202), anti-VSV-G-tag (GeneScript, #A00199), anti-TRAF3IP3 (Santa Cruz, #sc-366384), anti-cGAS (CST, #15102), anti-STING (CST, #13647), anti-TRAF3 (D1N5B, CST, #61095), anti-K48-linkage polyubiquitin (D9D5, CST, #8081). The following antibodies were used at 1:5000 dilution: anti-β-actin (Santa Cruz, #sc-1615-HRP conjugate), goat anti-mouse IgG light chain specific (Jackson Immunoresearch, #115-035-174), mouse anti-rabbit IgG light chain specific (Jackson Immunoresearch, #211-032-171). Proteins were detected using Femto Chemiluminescent reagent (Thermo Scientific, #34094). Densitometric measurements were made by Fiji ImageJ.

**Co-immunoprecipitation.** HEK293T cells were transfected with the indicated vectors for 24 h. THP-1 cells ($15 \times 10^6$ cells per time point) were infected with VSV infection for indicated times. Cells were washed with PBS and resuspended in lysis buffer (50 mM Tris-HCl, 150 mM NaCl, 1% NP-40 and 5 mM EDTA, protease inhibitor cocktail, phosphatase inhibitor cocktail) on ice for 30 min. Cell debris was pelleted by centrifugation at $16,000 \times g$ for 15 min at 4 °C. Immunoprecipitation was performed using anti-HA (EZview™ Red Anti-HA Affinity Gel, Sigma-Aldrich, #E6779), anti-FLAG (Anti-Flag M2 Affinity Gel, Sigma-Aldrich, #A2220), anti-Myc (EZview™ Red Anti-c-Myc Affinity Gel, Sigma-Aldrich, #E6654), anti-V5

(Anti-V5 Agarose Affinity Gel, Sigma-Aldrich, #A7345) agarose beads, or anti-TRAF3IP3 (Santa Cruz, #sc-366384) plus Protein G Sepharose 4 Fast Flow Beads (GE, #17061801) at 4 °C overnight. Then the beads were washed three times with lysis buffer. The immunoprecipitates were eluted using SDS loading buffer and denatured at 95 °C for 15 min. The eluted proteins were resolved by SDS-PAGE as described above.

**Ubiquitination assay**. For in vitro ubiquitination assays, HEK293T cells were transfected with the indicated vectors for 24 h. For endogenous ubiquitination assays, BMDMs ($8 \times 10^6$ cells) were mock infected or infected with VSV (MOI = 1) for indicated time. Cells were washed with PBS and resuspended in 1% SDS buffer (50 mM Tris-HCl, 150 mM NaCl, 1% NP-40 and 5 mM EDTA, protease inhibitor cocktail, phosphatase inhibitor cocktail). Lysates were boiled for 15 min for removal of any noncovalent interactions, then centrifuged for 10 min at $16,000 \times g$. The supernatants were transferred to a new tube and diluted 1:10 with lysis buffer (50 mM Tris-HCl, 150 mM NaCl, 1% NP-40 and 5 mM EDTA, protease inhibitor cocktail, phosphatase inhibitor cocktail). The diluted lysates were immunoprecipitated with anti-FLAG agarose beads (Anti-Flag M2 Affinity Gel, Sigma, #A2220), or anti-TBK1/NAK antibody (NOVUS, NB100-56705) plus Protein G Sepharose 4 Fast Flow Beads (GE, #17061801) overnight at 4 °C. After washing in the lysis buffer three times, the immunoprecipitates were subjected to SDS-PAGE as described above.

**Flow cytometry**. Cells were isolated from various organs harvested from 8-week-old sex-matched mice. Cells were blocked with anti-CD16/32 (Biolegend, #101301) (1:100 dilution), and then stained with the following antibody (1:200 dilution): BV711 anti-CD11b (Biolegend, #101241), BV785 anti-Ly-6G (Biolegend, #127645), AF700 anti-Ly-6C (Biolegend, #128023), APC anti-F4/80 (Biolegend, #123115), PE/Cy7 anti-CD11c (Biolegend, #117317), BV650 anti-CD19 (Biolegend, #115541), (Biolegend,), PE anti-CD3 (Biolegend, #100205), BV605™ anti-CD45.2 (Biolegend, #109841), and FITC anti-CD317 (Biolegend, #127007). All data were collected using FACSDiva 8.0 on Becton Dickinson LSRII (BD Biosciences) flow cytometer and analyzed using FlowJo V10 software (Tree Star, Ashland, OR).

**Flow sorting**. Cells were isolated from spleens and lymph nodes collected from 8-week-old sex-matched mice. Cells were blocked with anti-CD16/32 (Biolegend, #101301) (1:100 dilution) and then stained with the following antibody (1:200 dilution): FITC anti-CD19 (Biolegend, #152403), APC anti-CD11b (Biolegend, #101211) and PE anti-CD3 (Biolegend, #100205). The CD19+, CD3+, or CD11b+ cells were sorted using Becton Dickinson FACSAria II, rechecked by flow-cytometry to ensure purity at above 96%, then subjected to SDS-PAGE as described above.

**In vivo viral infection**. Eight-week-old sex matched mice were infected with VSV ($0.5 \times 10^7$ pfu of viruses per mouse) by intraperitoneal injection. The weight and survival of the mice were monitored accordingly. For cytokine studies, mice were sacrificed 8-h post infection and sera were collected through submandibular bleeding. For viral replication studies, mice were sacrificed 24-h post infection and spleen, lung, liver were dissected to determine viral titers.

**Statistical analyses**. Statistical analyses were carried out using Prism 7.0 (GraphPad, San Diego, CA). Data are presented as mean ± SEM or mean ± SD as denoted in Figure legends. Unpaired Student's $t$ tests were used for two-group analysis. If multiple subgroups were compared between two groups, the Holm-Sadak method was used for correction of the $p$ value. One-way ANOVA followed by Dunnett post hoc analysis or two-way ANOVA followed by Holm-Sidak post hoc analysis was used for multigroup comparisons. Log-rank (Mantel-Cox) test was used for survival analysis. In all tests, two-tail $p$ values or adjusted $p$ values of <0.05 were considered statistically significant, with $*p < 0.05$, $**p < 0.01$, $***p < 0.001$ as denoted.

**Metadata analysis**. For metadata analysis, human TRAF3IP3 expression was evaluated using the publicly accessible databases, including BioGPS[53] using dataset GeneAtlas U133A gcrma, Human Protein Atlas[54] using Tissue Atlas and using Cell Atlas, and Genecards[55] using 76 normal human tissues and compartments hybridized against Affymetrix Microarray HG-U133A. Mouse *Traf3ip3* expression was evaluated from the BioGPS using database GeneAtlas MOE430 gcrma. *Traf3ip3* mRNA measured by RNA-Seq in macrophage under VSV infection was from Infectome Map using Uniport ID Q8C0G2[38]. Hyperlinks to these data are provided in the Data availability section.

## Data availability
The following databases were used for metadata analysis. Human TRAF3IP3 expression was evaluated in BioGPS under Gene Atlas U133A gcrma, Human Protein Atlas with Tissue Atlas (TRAF3IP3) and Cell Atlas (TRAF3IP3), and GeneCards under the accession code GC01P209929 using 76 normal human tissues and compartments hybridized against Affymetrix Microarray HG-U133A. Mouse *Traf3ip3* expression was

evaluated in BioGPS using Gene Atlas MOE430 gcrma. We also used Infectome Map (http://www.infectome-map.org/) to analyze *Traf3ip3* mRNA expression in VSV infected macrophages using Uniport ID Q8C0G2. All other data supporting the findings of this study are available within the article and its supplementary information files and from the corresponding author upon reasonable request. The source data underlying Figures and Supplementary Figures are provided as a Source Data file.

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

## Acknowledgements

The UNC Flow Cytometry Core Facility and Microscopy Services Laboratory is supported by P30 CA016086 Cancer Center Core Support Grant to the UNC Lineberger Comprehensive Cancer Center. This work was supported by the US National Institute of Health (NIH) Grants U19-AI109965, R01-AI029564 to J.P.T., R01-DE026728 to Y.L.L., R15 GM134430-01 to B.K.D. J.W.T. was supported by NCI T32CA009156.

## Author contributions

M.D. and J.T. designed the study. M.D. and J.T. wrote the manuscript with input from J.W.T., W.J.B., B.K.D., S.S., C.C.K., and Y.L.L. M.D., J.W.T., L.Z., H.G., L.W., K.L., S.L., X.L. B.K.D., B.C., A.P., and Y.Z. performed the experiments and analyzed the data. S.S. provided *Traf3ip3*fl/fl mice. Y.L., S.S., C.C.K., and J.T. reviewed data. J.T. supervised the project.

## Competing interests

The authors declare no competing interests.
