## [Peer Review File · Nature Communications]

Reviewers' comments:

Reviewer #1 (Remarks to the Author):

TRAF3IP3 is an E3 ubiquitin ligase which is required for B and T cell development, as well as for the maintenance of regulatory T cell functional stability. In this study, Deng et al. found that TRAF3IP3 targeted TBK1 for K48-linked polyubiquitination and degradation, therefore, negatively regulated RLR-mediated innate response. This study is potentially interesting and contributes to our understanding of the molecular mechanisms of RLR-mediated innate immunity. The following points should be considered to improve the manuscript.

1. In Fig. 1a&b, the amounts of TRAF3IP3 were mistakenly labelled.
2. CoIP experiments for endogenous associations of TRAF3IP3 with TBK1 and TRAF3 should be performed.
3. Fig. 5h: the conclusion that TRAF3IP3 suppressed K63-linked polyubiquitination of TBK1 is not accurate, as the amount of TBK1 was decreased in the presence of TRAF3IP3. The effects of TRAF3IP3-deficiency on endogenous ubiquitination of TBK1 should also be examined.
4. It has been reported that some E3 ubiquitin ligases catalyze K48-linked polyubiquitination of TBK1 at Lys251, Lys372 and Lys670. Which residues in TBK1 does TRAF3IP3 targets for K48-linked polyubiquitination? As the authors stated, NLRP4-USP38, TRIP and Siglec1-TRIM27 have been reported to promote K48-linked polyubiquitination of TBK1. How do these components relate to TRAF3IP3 in promoting K48-linked polyubiquitination of TBK1?
5. What are the effects of TRAF3IP3-deficiency on innate immune response to DNA virus?

Reviewer #2 (Remarks to the Author):

Deng et.al., in this interesting and well written manuscript ascribe an IFN regulatory function to an already known protein TRAF3IP3 in RNA-induced anti-viral response. With the help of overexpression and knock out/knock down studies in 293T cells they show that Traf3Ip3 deficiency results in an increased IFN response and therefore allows for better viral clearance. In vivo experiments using flox Lyz2cre mice specifically targeting myeloid deletion of the protein demonstrates the negative regulation patterns of TRAF3IP3 resulting in better survival rates in mice infected with VSV.

MAJOR COMMENTS:

1. In figure 1j, the differences in phosphorylation of IRF3 induced in response to poly (dA:dT) and 5'PPP RNA in the presence or absence of TRAF3IP3 in only marginal. The authors must include IRF3 dimerization in these panels and TRAF3IP3 in 1i.
2. In Fig. 2c&d, the differences between control and knock-outs in phosphorylation of TBK1 and IRF3 are again only minor, although the densitometry analysis adds some value to the data, this is hardly sufficient. Additionally, if the induction of ISRE is higher with 5PPP, why not use that instead of PolyIC? Also, why do clones#1 and #2 behave similarly in spite of residual protein expression of TRAF3IP3 in clone#1? Visually, the p-IRF3 band at 3h in #1 seems more intense than #2 although the quantification in S4b (if this is the time point-this need to be specified in the legend) says otherwise. Efficiency of CRISPR/Cas9 knock out (for both clones, but #1 in particular) must be determined by indel frequency to establish a measure of successful knock out, especially since siRNA knock down in 4g looks to be just as efficient. It would also be nice to include TRAF3IP3 blots in Fig 2c&d.

3. Fig3a: Traf3ip3fl/fILyz2Cre+ myeloid cells are positive for TRAF3IP3? Has this been annotated incorrectly? This needs to be rectified.
4. Fig3 f&g: Contrary to what is stated, phosphorylation of 4-E-BP1 seems to be induced (at 6 and 9h) in the absence of TRAF3IP3 upon VSV infection, although this may not be the case in the context of Poly I:C transfection. Also, in both figures (3f&g) pERK expression is higher in the Cre+ cells at basal levels- how do the authors explain this?
5. Fig 5b: What is the rationale for looking at STING interaction with TRAF3IP3 in HEK cells? What induces binding of TRAF3IP3 with TBK1? If TBK1 needs to be modified, this experiment will require the presence of cGAS- (if transiently transfected?) Are TRAF3IP3 and TBK1 bound under normal/basal conditions?
6. What is the protein domain of TRAF3IP3 that interacts with TRAF3 and TBK1? Is there a mutant to this domain/domains that could serve as a negative control?
7. The role for ubiquitination in Fig 5h is difficult to appreciate given that the myc expression is unequal, which at least in the case of K63 could be the explanation for the observed decrease in Ub expression.
8. Fig 6 d-f, were the lung and the liver perfused before these measurements were made? Is this the effect of resident or infiltrating myeloid cells devoid of Traf3ip3?
9. The authors rely heavily on overexpression in in vitro studies performed on HEK cells. Can the authors use confocal imaging to show colocalisation of TRAF3IP3 with TBK1? –something more on endogenous protein?

MINOR COMMENTS:

1. Fig.1 (a-h) Is TRAF3IP3 transfected in a dose-dependent manner? If so, this is neither mentioned in the text nor is depicted in the figure (as shown in S2a).

We sincerely thank the reviewers for their constructive reviews. These comments helped us improve the quality of this study. We have carefully revised the manuscript with regard to the referees' comments. In doing so, we have included 13 panels of new data as suggested by reviewers and reorganized some sections.

Our responses to the referees' comments are noted in bold in the following pages. Most noteworthy, we have performed the very difficult experiment of showing the interaction of endogenous TRAF3IP3 and TBK1 in primary cells as requested by both reviewers. We further analyzed the effect of VSV on this interaction. Most important, we provide data using myeloid-specific deletion primary macrophages to show that endogenous TBK1 degradative K48 ubiquitination is dramatically reduced in the absence of *Traf3ip3* in VSV infected cells. This indicates that the protein is an attenuator of IFN response during RNA viral infection through K48-ubiquitination of TBK1. Furthermore, we identified the ubiquitination site by which TBK1 is ubiquitinated by TRAF3IP3 by site specific mutagenesis of putative ubiquitination sites. We also identified an E3 ligase, DTX4, that is important in TBK1 ubiquitination by TRAF3IP3 and defined the main domains which mediate TBK1 and TRAF3IP3 interaction. In sum, this is the first report of a function for TRAF3IP3 in the myeloid lineage. It is supported by thorough biochemical data, many of which are performed with primary cells on endogenous proteins, to indicate a clear pathway that leads to physiologic outcomes that affect an anti-viral response.

Due to these greatly expanded data, we have modified the title and rewrote the abstract.

All changes in the main text are marked by blue lines in the left margin.

Reviewer #1 (Remarks to the Author):

TRAF3IP3 is an E3 ubiquitin ligase which is required for B and T cell development, as well as for the maintenance of regulatory T cell functional stability. In this study, Deng et al. found that TRAF3IP3 targeted TBK1 for K48-linked polyubiquitination and degradation, therefore, negatively regulated RLR-mediated innate response. This study is potentially interesting and contributes to our understanding of the molecular mechanisms of RLR-mediated innate immunity.

We thank the reviewer for these very encouraging comments regarding the strengths of the work.

1. In Fig. 1a&b, the amounts of TRAF3IP3 were mistakenly labelled.

We apologize for this mistake. We relabeled the TRAF3IP3 in wedge indicating increasing transfection dose. We also indicated in the legend the doses used in the legend.

2. CoIP experiments for endogenous associations of TRAF3IP3 with TBK1 and TRAF3 should be performed.

These experiments to investigate the endogenous association of TRAF3IP3 with TBK1 and TRAF3 are very challenging. However after numerous attempts, we have been able to optimize the conditions and obtain endogenous co-IP results. In addition, we show that the interaction of TRAF3IP3 with TBK1 and TRF3 are increased upon VSV infection. This provides important validation for the results obtained with over-expressed protein. The result is included as Fig. 5e.

3. Fig. 5h: the conclusion that TRAF3IP3 suppressed K63-linked polyubiquitination of TBK1 is not accurate, as the amount of TBK1 was decreased in the presence of TRAF3IP3. The effects of TRAF3IP3-deficiency on endogenous ubiquitination of TBK1 should also be examined.

We thank the reviewer for pointing this out. We agree and revised the text by stating “TRAF3IP3 specifically increased K48-linked but not K63-linked polyubiquitination of TBK1”.

Regarding the effects of TRAF3IP3-deficiency on endogenous ubiquitination of TBK1, we agree with the reviewer that this is an important issue. However the test of endogenous TBK1 ubiquitination resulting in high quality western blots showing K48 Ub has been extremely challenging. Nonetheless, after multiple tries we have obtained clear data regarding the endogenous ubiquitination of TBK1 in the *Traf3ip3^{fl/fl} Lyz2-cre* cells. The result shows very strong K48 ubiquitination of TBK1 in wildtype cells infected with VSV, but not in cells with myeloid specific deletion of *Traf3ip3*. This important piece of data is included in Supplementary Fig. 7a.

4. It has been reported that some E3 ubiquitin ligases catalyze K48-linked polyubiquitination of TBK1 at Lys251, Lys372 and Lys670. Which residues in TBK1 does TRAF3IP3 targets for K48-linked polyubiquitination?

To address this fully, we have made constructs containing several mutations mentioned in TBK1 (Lys82, Lys241, Lys251, Lys 341, Lys372). We show that K372R is the critical residue (Fig. 6e).

As the authors stated, NLRP4-USP38-DTX4, TRIP and Siglec1-TRIM27 have been reported to promote K48-linked polyubiquitination of TBK1. How do these components relate to TRAF3IP3 in promoting K48-linked polyubiquitination of TBK1?

We produced two shRNAs each for DTX4, TRIP aand TRIM27. The outcome shows that sh-DTX4 interfered with TRAF3IP3 activated K48 ubiquitination. The result is included in Fig. 6d, with Supplementary Fig. 7b showing the efficacy of shRNA gene reduction. We mentioned the possible involvement of USP38/DTX4 in the abstract and the discussion.

5. What are the effects of TRAF3IP3-deficiency on innate immune response to DNA virus?

We are just starting to investigate TRAF3IP3-deficiency on innate immune response to DNA virus. However, our data are not straightforward, and a thorough investigation of that observation will likely take another year and is beyond the scope of this paper.

Reviewer #2 (Remarks to the Author):

Deng et.al., in this interesting and well written manuscript ascribe an IFN regulatory function to an already known protein TRAF3IP3 in RNA-induced anti-viral response. With the help of overexpression and knock out/knock down studies in 293T cells they show that Traf3Ip3 deficiency results in an increased IFN response and therefore allows for better viral clearance.

In vivo experiments using flox Lyz2cre mice specifically targeting myeloid deletion of the protein demonstrates the negative regulation patterns of TRAF3IP3 resulting in better survival rates in mice infected with VSV.

We thank the reviewer for these exceptionally complementary and encouraging comments.

MAJOR COMMENTS:

1. In figure 1j, the differences in phosphorylation of IRF3 induced in response to poly(dA:dT) and 5'PPP RNA in the presence or absence of TRAF3IP3 in only marginal.

We re-performed the experiment using a different time point and present the results in Fig. 1j and Supplementary Fig. 2d. This earlier time point showed a modest increase of the difference. We agree that the differences we measured in in vitro cell line for IRF3 or TBK1 are typically in the two-fold or less range. However when we bred the *Traf3ip3^{fl/fl}* Lyz2-Cre mice to assay specific impact of myeloid deletion of Traf3ip3, the differences in the IFN signaling pathway between control and knockouts showed a clear pattern of difference (see Fig. 3) and the in vivo impact on viral infection is also significant (Fig. 7). The data obtained with primary cells or mice are more physiologically relevant and hence we believe should provide more weight in supporting the model that TRAF3IP3 reduces a number of signaling pathways.

2. In Fig. 2c&d, the differences between control and knock-outs in phosphorylation of TBK1 and IRF3 are again only minor, although the densitometry analysis adds some value to the data, this is hardly sufficient. Additionally, if the induction of ISRE is higher with 5PPP, why not use that instead of PolyIC? Also, why do clones#1 and #2 behave similarly in spite of residual protein expression of TRAF3IP3 in clone#1?

We selected poly(I:C) because initially we were looking for a function for Traf3ip3 in myeloid cells and we were testing different pathways including RNA sensing pathways and used poly(I:C) to cast a wider net. As a result most of the followup experiments used poly(IC) (Fig. 2c, e, f, g; Fig. 3b, c, f). The inclusion of

5'ppp-dsRNA was added later. Both poly(I:C) and 5'ppp-dsRNA resulted in high ISRE- and IFN- β luc reporter activation (Fig. 1a vs. 1e, 1b vs. 1f). If we switch to 5'ppp-dsRNA, the followup experiments in Figures 2 and 3 would have to be repeated.

In fig. 2b, the difference between the #1 and #2 sgRNA is modest, and only visible upon a long exposure. Additionally, the #2 clone did produce enhanced pTBK1 in poly(IC) stimulation and higher responses compared to #1 clone in Fig. 2e through 2j (except for IFN β 1 mRNA at the 3 hour time point). In multiple cases, the #2 clone produced greater significance values than the #1 clone when compared to control (see Fig. 2e 6h, 2f, 2h 6 hr, 2j).

Visually, the p-IRF3 band at 3h in #1 seems more intense than #2 although the quantification in S4b (if this is the time point-this need to be specified in the legend) says otherwise.

We double-checked densitometry and it is correct. We performed the densitometry of p-IRF3 against β -actin, and the densitometry of p-IRF3 is the sum of intensity of p-IRF3 of one clone at 0, 3, 6 and 9 h lane against the sum of the corresponding β -actin intensity.

Efficiency of CRISPR/Cas9 knock out (for both clones, but #1 in particular) must be determined by indel frequency to establish a measure of successful knock out, especially since siRNA knock down in 4g looks to be just as efficient. It would also be nice to include TRAF3IP3 blots in Fig 2c&d.

We have determined the indel frequency. We have attached the result in Supplemental Fig. 4a. We have also included TRAF3IP3 blots in Fig. 2c, d.

3. Fig3a: Traf3ip3fl/flLyz2Cre+ myeloid cells are positive for TRF3IP3? Has this been annotated incorrectly? This needs to be rectified.

We apologize for this mistake. We corrected the mistake. CD11b+ myeloid cells from Traf3ip3fl/flLyz2Cre+ mice are negative for TRAF3IP3 as expected. By contrast, cells without Lyz2Cre (the – lanes) still have TRAF3IP3 protein.

4. Fig3 f&g: Contrary to what is stated, phosphorylation of 4-E-BP1 seems to be induced (at 6 and 9h) in the absence of TRAF3IP3 upon VSV infection, although this may not be the case in the context of Poly I:C transfection. Also, in both figures (3f&g) pERK expression is higher in the Cre+ cells at basal levels- how do the authors explain this?

We thank reviewer for pointing this out. We have revised our description of these results in the manuscript (also see Supplementary Fig. 6a and b for densitometry). We also noted that the most dramatic and consistent difference in both 3f and g is enhanced p-p38 in both Poly I:C and VSV infected cells. p38 is part of the IFN signaling pathway after type I IFN engagement with the receptor (Platanias LC, Pharmacol Ther. 98:129; Uddin S et al., J. Biol Chem 274:30127).

This would be consistent with a working hypothesis that TRAF3IP3 promotes TBK ubiquitination leading to reduced IFN which then leads to reduced p38 phosphorylation. Deletion of Traf3ip3 results in enhanced TBK as well as p38, but TBK1 lies upstream of IFN production while p38 lies downstream of IFN production.

5. Fig 5b: What is the rationale for looking at STING interaction with TRAF3IP3 in HEK cells?

We used STING as an additional control but after the reviewer's comment, we agree that it is not well rationalized, and thus we have removed the STING interaction panel.

What induces binding of TRAF3IP3 with TBK1? ...Are TRAF3IP3 and TBK1 bound under normal/basal conditions?

We agree that this is an important experiment and should be performed with endogenous protein at the basal/normal conditions. However the experiments to investigate the endogenous association of TRAF3IP3 with TBK1 are very challenging. After numerous attempts, we have been able to optimize the conditions and obtain endogenous co-IP results. We show that the interactions of TRAF3IP3 with TBK1 and TRF3 are minimal at the baseline level, but is increased upon VSV infection. This provides important validation for the results obtained with over-expressed protein. The result is included as Fig. 5e. In addition, we also examined TRAF3IP3 mRNA level after Poly(I:C) transfection or VSV infection and attached the data in Supplementary Fig. 4h, i, j. The data show that Poly(I:C) and VSV do not alter the level of Traf3ip3 transcript.

6. What is the protein domain of TRAF3IP3 that interacts with TRAF3 and TBK1?

We have constructed a number of mutants and performed the reciprocal domain mapping co-IP of TRAF3IP3 and TBK1 to address this issue. The results are shown in 5b, c. The domain of TBK1 that interacts with TRAF3IP3 is in the residue 1-300 KD domain. We truncated TRAF3IP3, the results showed that 1-235 and 455-551 residues have the strongest interacting sites (Supplemental Fig. 7b). The domain of TRAF3IP3 interacting with TRAF3 was already previously reported in early paper "T3JAM, a novel protein that specifically interacts with TRAF3 and promotes the activation of JNK(1). FEBS Lett. 2003".

7. The role for ubiquitination in Fig 5h is difficult to appreciate given that the myc expression is unequal, which at least in the case of K63 could be the explanation for the observed decrease in Ub expression.

We reorganized the Fig. 5g to Fig. 6C. We agree. We revised the text by stating "TRAF3IP3 specifically increased K48-linked but not K63-linked polyubiquitination of TBK1".

8. Fig 6 d-f, were the lung and the liver perfused before these measurements were made? Is this the effect of resident or infiltrating myeloid cells devoid of Traf3ip3?

We did not perfuse the lung and the liver before these measurements were made thus we do not make any claim of specific effects on resident or infiltrating myeloid cells devoid of Traf3ip3.

9. The authors rely heavily on overexpression in in vitro studies performed on HEK cells. Can the authors use confocal imaging to show colocalisation of TRAF3IP3 with TBK1? –something more on endogenous protein?

We agree with this critique. We have tried several TBK1 antibodies in multiple experiments, unfortunately these antibodies gave false positive staining in the cell nucleus. We are not confident about the data thus we prefer not to include the data.

However we have used biochemical approaches to address the role of the endogenous protein. First, we performed co-IP experiments assessing the endogenous associations of TRAF3IP3 with TBK1 and TRAF3. We have been able to optimize the conditions and obtain endogenous co-IP results that is publication quality. In addition, we show that the interaction of TRAF3IP3 with TBK1 and TRF3 are increased upon VSV infection. This provides important validation for the results obtained with over-expressed protein, that TRAF3IP3 is important for the resolution of an IFN response. The result is included as Fig. 5e.

Second, we tested the endogenous effect of TRAF3IP3 on TBK1 ubiquitination in primary cells lacking TRAF3IP3 specifically in the myeloid lineage. The testing of TRAF3IP3 on endogenous K48 Ub of TBK1 is extremely challenging, however after numerous trials, we succeeded in generating high quality western blots showing K48 Ub of TBK1 is enhanced by TRAF3IP3. We believe these are clear data showing that the endogenous ubiquitination of TBK1 is greatly reduced in the Traf3ip3fl/fl x Lyx2-cre cells as indicated by very strong K48 ubiquitination of TBK1 in wildtype cells infected with VSV, but not in cells with myeloid specific deletion of Traf3ip3. This important piece of data is included in Supplementary Fig. 7b.

MINOR COMMENTS:

1. Fig.1 (a-h) Is TRAF3IP3 transfected in a dose-dependent manner? If so, this is neither mentioned in the text nor is depicted in the figure (as shown in S2a).

We apologize for this omission. We relabeled the TRAF3IP3 in wedge indicating increasing transfection dose. We also indicate the concentration of the wedge in the legend.